# Soil respiration of alpine meadow is controlled by freeze-thaw processes of active layer in the permafrost region of the Qinghai-Tibet Plateau

Junfeng Wang[1, 2], Qingbai Wu[1], Ziqiang Yuan[1], Hojeong Kang[3]

[1] State Key Laboratory of Frozen Soil Engineering, Northwest Institute of Eco-Environment and Resources, CAS, Lanzhou 730000, China
[2] Beiluhe Observation Station of Frozen Soil Environment and Engineering, Northwest Institute of Eco-environment and Resources, CAS, Lanzhou 730000, China
[3] School of Civil and Environmental Engineering, Yonsei University, Seoul 03722, Korea

*Correspondence to*: Hojeong Kang (hj_kang@yonsei.ac.kr), Qingbai Wu (qbwu@lzb.ac.cn)

**Abstract:** Freezing and thawing action of the active layer plays a significant role in soil respiration ($R_s$) in permafrost regions. However, little is known about how the freeze-thaw processes affect the $R_s$ dynamics in different stages of the alpine meadow underlain by permafrost in the Qinghai-Tibet Plateau (QTP). We conducted continuous *in-situ* measurements of $R_s$ and freeze-thaw process of the active layer at an alpine meadow site in the Beiluhe permafrost region of QTP, and divided the freeze-thaw processes into four different stages in a complete freeze-thaw cycle, including summer thawing stage (ST), autumn freezing stage (AF), winter cooling stage (WC), and spring warming stage (SW). We found that the freeze-thaw processes have various effects on the $R_s$ dynamics in different freeze-thaw stages. The mean $R_s$ ranged from 0.12 to 3.18 μmol/m²s across the stages, with the lowest value in WC and highest value in ST. $Q_{10}$ among the different freeze-thaw stages changed greatly, with the maximum (4.91±0.35) in WC and minimum (0.33±0.21) in AF. Patterns of $R_s$ among the ST, AF, WC, and SW stages differed, and the corresponding contribution percentages of cumulative $R_s$ to total $R_s$ of a complete freeze-thaw cycle (1692.98±51.43 gCO₂/m²) were 61.32±0.32, 8.89±0.18, 18.43±0.11, and 11.29±0.11%, respectively. Soil temperature ($T_s$) was the most important driver of $R_s$ regardless of soil water status in all stages. Our results suggest that as climate change and permafrost degradation continue, great changes in freeze-thaw process patterns may trigger more $R_s$ emissions from this ecosystem because of prolonged ST stage.

**Keywords:** Soil respiration; Different freeze-thaw stage; $Q_{10}$; Alpine meadow; Qinghai-Tibet Plateau

## 1 Introduction

Soil respiration ($R_s$) is a significant source in estimating terrestrial carbon budget under climate change. It is the second-largest source of carbon emissions to the atmosphere from the terrestrial ecosystem on a global scale (Bond-Lamberty and Thomson, 2010;Schlesinger and Andrews, 2000). In permafrost regions, $R_s$ not only depends on the distribution of vegetation and the content of soil organic matter (SOM) (Ping et al., 2008;Grogan and Chapin III, 2000;Phillips et al., 2011;Jobbágy and Jackson, 2000), but also is regulated by the freeze-thaw process of active layer (Hollesen et al., 2011). Many studies have shown that the winter-time emissions contribute significantly to the

annual $CO_2$ balances. For example, the Arctic tundra ecosystem is becoming a consistent source of $CO_2$ because $CO_2$ emission in winter offsets its uptake in growing season with progressive permafrost thaw and active layer thickening (Celis et al., 2017). In Alaska, emissions of $CO_2$ from tundra during early winter seasons increased by about 73% since 1975, and the Arctic ecosystem has been a net source of $CO_2$ due to rising temperatures (Commane et al., 2017). For the sub-arctic tundra ecosystem, the winter-time $CO_2$ loss has also been increasing due to sustained tundra warming, and as a result the ecosystem's historical function is shifting away from a carbon sink to a carbon source (Lüers et al., 2014;Webb et al., 2016). In permafrost regions in the northern hemisphere, the amount of soil organic carbon (SOC) stored reaches 1832Pg (Ding et al., 2015;Tarnocai, 2009), among which about 689 Pg distributes in the 0-1 m depth, 1035±150 Pg in the 0-3 m depth and 648 Pg in the 3-25 m depth (Hugelius et al., 2014;Tarnocai et al., 2009). Due to its high sensitivity to global warming and direct contribution to the atmosphere greenhouse gas contents, carbon emission from permafrost regions has received worldwide attention (Tarnocai, 2009;Zimov et al., 2009).

Both active layer and whole permafrost distributed in the Arctic and mid-latitude alpine regions are undergoing significant changes due to global warming (Jorgenson and Osterkamp, 2005). The active layer, which acts as a buffer between permafrost and atmosphere, is highly sensitive and responsive to climate change (Li et al., 2012). The exchange of energy and water in permafrost regions between the land and the atmosphere mainly occurs through the active layer. However, in a whole freeze-thaw cycle, the active layer will undergo a series of cooling, start freezing to fully freezing, dropping in temperature, rising in temperature but still in frozen state, start thawing to fully thawing, and rising in temperature but in thawed state (Jiao and Li, 2014). At different developing stages of freeze-thaw cycling, the heat distribution and transmission in the active layer show significantly different characteristics (Zhao et al., 2000). Thus the soil physicochemical properties, microbial activities, and biogeochemical processes at different freeze-thaw stages are also different from each other (Henry, 2007). As such, the dynamics of $R_s$ emission at different freeze-thaw stages may show apparent differences. Furthermore, the thawing of permafrost and the deepening of the active layer will expose frozen organic carbon to microbial decomposition, and cause the previously frozen SOC to become available for mineralization (Walz et al., 2017). This may accelerate a positive permafrost carbon feedback to climate change (Schuur et al., 2008). A six-year study of $CO_2$ flux in moist acidic tundra has shown that the active layer thickness is a key driver of NEE, GPP, and ecosystem respiration (Celis et al., 2017). In high-altitude mountain regions, permafrost thawing has caused the alpine tundra to release $CO_2$ from organic carbon stored for a long time to the atmosphere, exacerbating climate change (Knowles et al., 2019). Therefore, permafrost must be playing a significant role in carbon-climate feedbacks due to its intensity of climate forcing and its size of the carbon pools (MacDougall et al., 2012;Schneider von Deimling et al., 2012).

The strength and timing of permafrost carbon feedback essentially depend on the freeze-thaw process of the active layer and the distribution of SOC in permafrost regions. Therefore, understanding the effects of freeze-thaw actions on $R_s$ at different freeze-thaw stages is critical for better predicting future climate changes. However, it is still unclear how the freeze-thaw actions at different stages regulate the $R_s$.

The Qinghai-Tibet Plateau (QTP) of China has the largest extent of permafrost in the low-middle latitudes of the world and is very sensitive to global climate change (Liu and Chen, 2000;Wu et al., 2010). Soil organic carbon (SOC) pools in the permafrost regions of QTP were estimated to

be 160±87 Pg, which is approximately 8.7% of those in the northern circumpolar permafrost region (Mu et al., 2015). Recent years have witnessed dramatic changes in the freeze-thaw occurrence, active-layer thickness, and near-surface permafrost temperature in QTP. In the permafrost regions distributed with alpine meadow ecosystem in QTP between 2002 and 2012, the average onset of spring thawing at 50-cm depth advanced by at least 16 days; the duration of thaw increased by at least 14 days; the active-layer thickness increased by ~4.26 cm/a, and the near-surface permafrost temperature at 6 m and 10 m depths increased by ~0.13 °C and ~0.14 °C, respectively (Wu et al., 2015). Therefore, the $R_s$ of the alpine meadow is anticipated to be influenced and changed dramatically due to the variations of freeze-thaw occurrence, active-layer thickness, and near-surface permafrost temperature.

We took *in-situ* measurements of $R_s$ and freeze-thaw process of the active layer in an alpine meadow from January 2017 to December 2018. The objectives were (1) to determine the dynamics of the $R_s$ during a complete freeze-thaw process of active layer; (2) to compare the $R_s$ patterns among the different freeze-thaw stages and their contribution to total $R_s$ emission in a complete freeze-thaw cycle in this region; and (3) to establish a preferable $R_s$ model to accurately predict the soil $CO_2$ emission of each freeze-thaw stages.

# 2 Materials and methods

## 2.1 Study site

The experiment was conducted in an alpine meadow ecosystem of the Beiluhe region (34° 49′ 25.8″ N,92° 55′ 45.1″ E), in the hinterland of the QTP, China. The study site represents an area of 151.6 km$^2$, with an altitude of 4,600 – 4,800 m, which is underlain by continuous permafrost with an active layer of 1.1-2.3 m. The soil types in the study site are primarily classified as MatticGelic Cambisols (alpine meadow soil) in Chinese taxonomy or as Cambisols in FAO/UNESCO taxonomy (Wang et al., 2014). The mean annual temperature is -3.60 °C, which is lower than that of most of other areas in the QTP (Yin et al., 2017). The mean annual precipitation is 423.79 mm, 80% of which falls as rain, sometimes mixed with small hails during the growing season (from May to September). In winter, little snow falls but is quickly blown away and sublimated off due to high wind and low air temperature, so the study site is not persistently covered by snow. The air pressure is approximately 550 hPa. The alpine meadow represents the most common vegetation type in QTP, which cover more than 70% of whole area (Wang and Wu, 2013;Zhang et al., 2015b). The alpine meadow ecosystem mainly consists of cold meso-perennial herbs that grow in conditions where a moderate amount of water is available, such as *Kobresia pygmaea* (C. B. Clarke), *Kobresia humilis* (C. A. Meyer ex Trautvetter) *Sergievskaja*, *Kobresia capillifolia* (Decaisne) (C. B. Clarke), *Kobresia myosuroides* (Villars) Fiori, *Kobresia graminifolia* (C. B. Clarke), *Carex atrofusca Schkuhr subsp.* (minor (Boott) T. Koyama), and *Carex scabriostris* (Kukenthal) (Chen et al., 2017). On-site surveying and sampling of the experiment set-up showed that soil bulk density, soil organic carbon, and total N content at the 10-20 cm depth were higher than those at the 0-10 cm depth. The depth of the active layer was about 1.9 m. The belowground biomass was much greater than that of aboveground. The average depth of vegetation main rooting zone was around 10 cm (Table 1).

 Table 1 Biomass and soil properties at the experiment set-up

| Chemical and biological characteristics | Depth (cm) | Values |
|---|---|---|
| Bulk density (g cm$^{-3}$) | 0–10 | 0.89 ± 0.2 |
| | 10–20 | 0.98 ± 0.1 |
| Soil organic C (kg m$^{-2}$) | 0–10 | 0.48 ± 0.06 |
| | 10–20 | 1.32 ± 0.04 |
| Soil total N (g m$^{-2}$) | 0–10 | 41.3 ± 7.2 |
| | 10–20 | 117.6 ± 12.8 |
| Above-ground biomass (kg m$^{-2}$) | | 0.33 ± 0.04 |
| Below-ground biomass (kg m$^{-2}$) | | 2.41 ± 0.4 |
| Depth of vegetation main rooting zone (cm) | | 10 ± 3 |
| Active layer depth (m) | | 1.90 ± 0.2 |

Values are means ($n$ = 5) ± standard deviation (SD)

## 2.2 Measurement of the freeze-thaw process of the active layer

In the study site, one flat terrain with vegetation coverage of above 70% was selected to establish the active layer observation site. According to active layer lithology and practical conditions, soil temperature and soil moisture probes were installed at different depths. The installation depths for the soil temperature probes were 5, 20, 50, 80, 120, 150, 180 and 230 cm, and the depths for the soil moisture probes were 5, 20, 50, 80, 120, 150 and 180 cm. Soil temperature was measured using thermistors made by the State Key Laboratory of Frozen Soil Engineering (SKLFSE, China) with the accuracy of ±0.05°C. Soil moisture was measured using calibrated soil moisture sensors (EC-5, Decagon USA) with the accuracy of ±0.02 m$^3$m$^{-3}$. Soil moisture measured using the EC-5 probe represents the volumetric water content of liquid water per total soil volume. These measurements were collected automatically every 30 min each day by a data logger (CR3000, Campbell Co., USA).

Utilizing the measurements collected by soil temperature probes and a data logger, soil hourly mean temperature ($T_{avg}$), maximum temperature ($T_{max}$), and minimum temperature ($T_{min}$) of each day at different depths were calculated. Assuming that the soil particle surface energy and the salinity of soil having no influence on the soil freezing temperature (Jiao and Li, 2014), the date on which hourly $T_{avg}$ continued to be lower or higher than 0 °C was regarded as the onset freezing or onset thawing date, respectively, according to the $T_{avg}$ values (Yang et al., 2002). If $T_{max}$ was greater than 0 °C and $T_{min}$ was less than 0 °C in a single day, it was regarded that the soil was undergoing daily freeze-thaw process. That is, the soil absorbs heat and thaws during the daytime, and releases heat and freezes during the nighttime, showing a daily freeze-thaw cycling phenomenon. Based on these criteria, whole freeze-thaw process of active layer can be divided into different stages.

To calculate the freezing or thawing thickness of the active layer in the freeze-thaw process, freeze or thaw depth was estimated by linearly interpolating soil temperature profiles between two neighboring points above and below the 0°C isotherm (Wu et al., 2010). The freezing or thawing thickness of the active layer was estimated from daily soil temperature measurements.

## 2.3 Soil respiration measurement

For the measurements of $R_s$, six 5×5m plots were randomly selected around the active layer observation site, and one polyvinyl chloride (PVC) collar (20 cm in internal diameter and 10 cm in

height) was inserted into each plot to a depth of 8 cm into the soil with a chamber offset of 2 cm before the soil froze. All the PVC collars were left in place until the end of the study. $R_s$ flux was measured using an LI-8100A automated soil gas flux system (LI-COR Inc., Lincoln, NE, USA). A standard LI-COR® 20-cm head was applied for measurements with 10-ppm range set and the offset of the program was adjusted to 2 cm. A typical survey measurement protocol was adopted with an observation length of 2 minutes, deadband of 25 seconds, pre-purge of 30 seconds and post-purge of 45 seconds. The chamber volume and the IRGA volume were automatically calculated by the program. Living plants inside the collar were removed carefully on the soil surface at least one day before the measurement. $R_s$ flux was measured for two years covering a complete freeze-thaw cycle of the active layer between 2017 and 2018.

$R_s$ flux was determined once every two or three days during the thawing period and once every seven days during the freezing period due to harsh environmental conditions and lack of manpower. Measurements were taken between 9:00 and 11:30 a.m. local time on every sampling day to represent the daily average flux based on the diurnal measurements (Zhang et al., 2015a). At the same time, soil temperature ($T_s$) and soil volumetric water content (SWC) at 5cm depth were determined besides the collars using the thermocouple probe and the ECH2O soil moisture sensor (LI-COR, Lincoln, NE, USA) connected to the LI-8100A.

## 2.4 Temperature sensitivity and scaling for $R_s$ at different freeze-thaw stage

For each freeze-thaw stages, the relationship between $R_s$ flux and soil temperature or soil water content was determined by fitting to exponential and polynomial functions given in equations (1) and (2), respectively (Zhang et al., 2015a).

$$R_S = \beta_0 e^{\beta_1 T} \qquad (1)$$
$$R_S = aSWC^2 + bSWC + c \qquad (2)$$

where $R_s$ is the measured soil respiration rate ($\mu mol\, m^{-2} s^{-1}$); $T$ and SWC are soil temperature and water content at 5cm depth, respectively; $\beta_0$, $\beta_1$, a, b and c are coefficients. The exponential relationship is commonly used to represent soil respiration and soil carbon efflux as functions of temperature (Janssens and Pilegaard, 2003;Davidson et al., 1998). $Q_{10}$ represents the temperature sensitivity of $R_s$, which is a measure of change in reaction rate at intervals of 10°C and is based on Van't Hoff's empirical rule (Lloyd and Taylor, 1994). $Q_{10}$ based on Eq. (1) was calculated as Eq. (3) (Davidson and Janssens, 2006;Davidson et al., 1998).

$$Q_{10} = e^{10\beta_1} \qquad (3)$$

The daily average $R_s$ flux for the different freeze-thaw stages was obtained based on the best fitting equations (1) or (2) and the corresponding daily average soil temperatures or soil water contents at 5cm depth measured by a data logger set up at the study site. The cumulative $R_s$ emission at the different freeze-thaw stage was calculated by computing the sum of products of the average flux rate and the start-stop-time of the different freeze-thaw stage of the active layer as follows (Zhang et al., 2017),

$$SR = \sum_{k}^{m} R_{mk} \qquad (4)$$

where k and m are the corresponding onset date and end date of each freeze-thaw stage, respectively; $R_{mk}$ is the daily $R_s$ emission over the specific freeze-thaw stage.

## 2.5 Statistical analysis

Repeated measures ANOVA was applied for testing the statistical significance of the differences among freeze-thaw stages. Regression analysis was performed between $R_s$ and soil

temperature and soil moisture. At different freeze-thaw stages, the fitting equations with higher $R^2$ values were selected as the preferable models to predict the daily soil $CO_2$ emission. To evaluate the reliability of the $R_s$ models at the different freeze-thaw stages, root mean squared error ($RMSE$) analysis was performed. All statistical analyses were carried out at a significance level of 0.05 and were completed using SPSS 16.0 (SPSS Inc., Chicago, IL, USA).

## 3 Results

### 3.1 Division of different freeze-thaw stages of the active layer

We assumed that the soils began to freeze when the temperature dropped lower than 0°C, and to thaw when the temperature was continuously greater than 0°C, based on the fact that the effects of surface energy of soil particles and salinity in soil on freezing temperature are negligible. Two years' continuous observation on the freezing and thawing of the active layer in the study site showed that the contour outline of 0°C began to slowly develop downwards from the soil surface from late April, and reached the maximum depth in the early October (Fig. 1). The maximum thawing depth of the active layer was 1.98 m in 2017 and 1.89 m in 2018. During this thawing period, the isotherm of 0°C changed gently. However, the isotherm of 0°C changed rapidly from early October to late November, indicating that the whole active layer froze from the surface to the bottom in a short period of time. According to the variations in soil temperature and soil water content in the active layer, the freezing and thawing cycle process of the active layer was divided into four distinctive stages; summer thawing stage (ST), autumn freezing stage (AF), winter cooling stage (WC), and spring warming stage (SW).

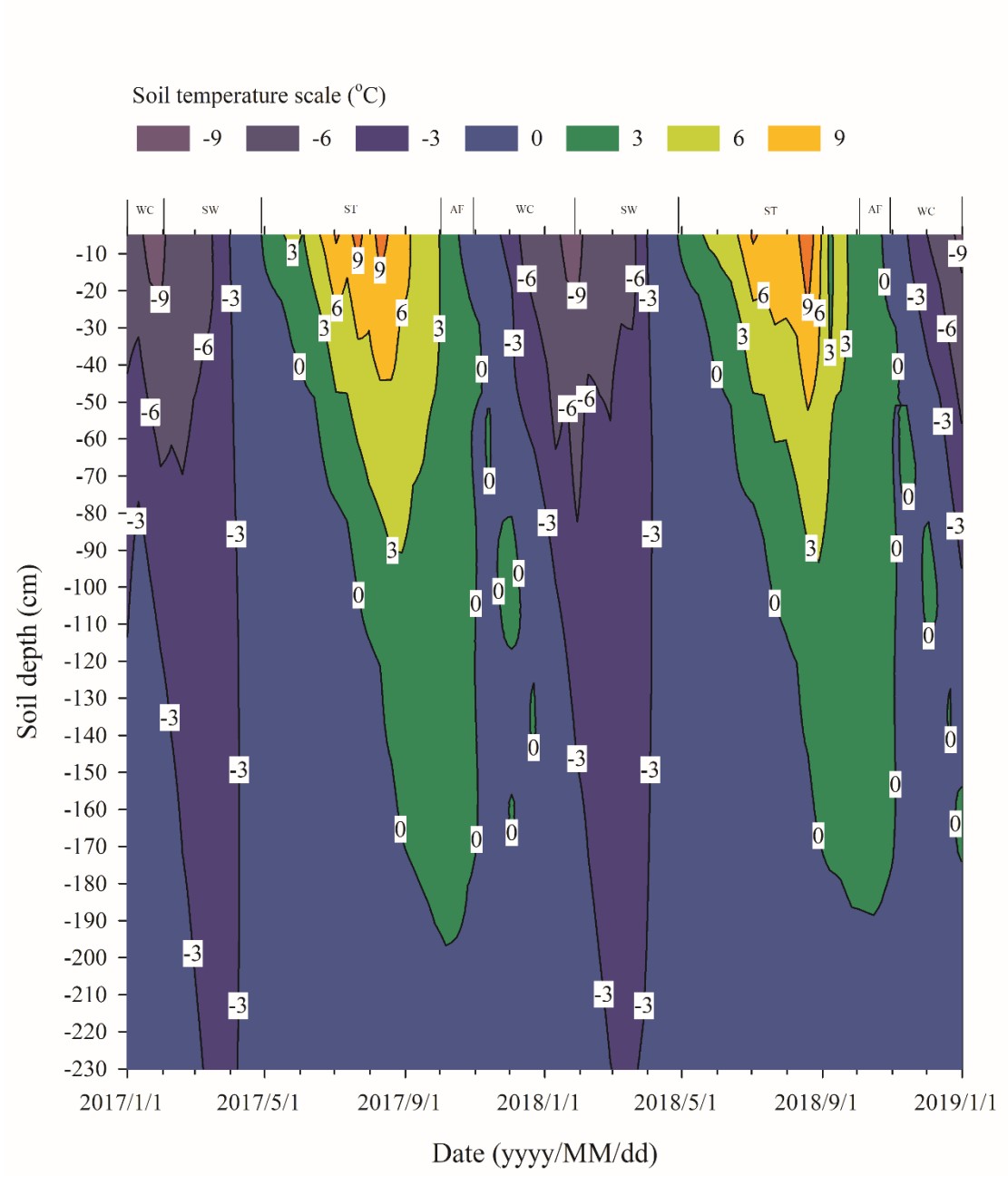

Fig.1. Soil temperature contour outlines of the experimental site in 2017 and 2018

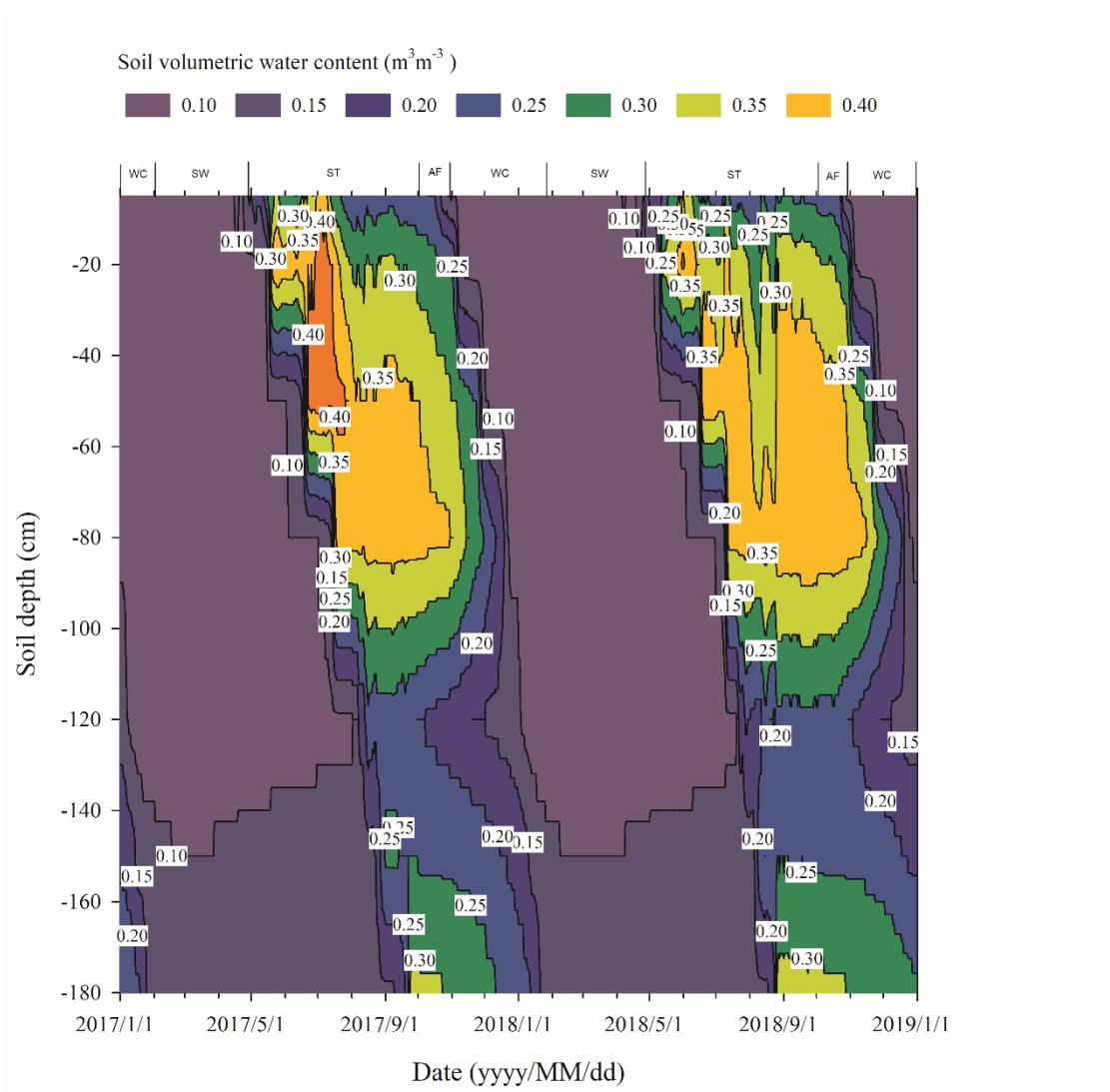

Fig.2. Soil moisture contour outlines of the experimental site in 2017 and 2018

The process of ST stage started when the active layer began to thaw downwards from the surface in late April, and preceded to early October when the thawing depth reached the maximum. At this stage, soil temperature decreased along with the soil depth and soil water was mainly transported downward. The whole active layer was in an endothermic process where the heat transferred downwards continuously and the soil thawing front also slowly migrated downwards.

At AF stage, once the thawing depth reached the maximum, the soil began to freeze upwards from the bottom of the active layer. Thus, AF lasted until the whole active layer was frozen. At AF stage, soil water, being driven by the temperature gradient, migrated to both sides of the freezing front from the thawing layer and froze there. During AF, the slope of 0°C isotherm was flat. Especially, 0°C isotherm almost paralleled the axis of ordinate between the depth of 50 cm and 160 cm. This

phenomenon appeared because the onsets of freezing at different depths had no apparent differences and the whole active layer completed the freezing process in a short time as AF process started. Once the whole freezing process was completed, WC stage quickly started and lasted until mid-late January of the next year. During this process, the soil temperatures were relatively lower in the upper active layer but higher at the bottom; a small amount of soil water near the surface evaporated and

the unfrozen water in the active layer tended to migrate upward (Fig.2). However, the amount of soil water migration was small because the lower ground temperature limited the content and activity of the unfrozen water. The SW stage began in late January as the air-temperature rose, and temperature gradients in the active layer gradually decreased. During the SW stage, surface soil usually underwent daily freezing and thawing cycles in late April. The amount of soil water evaporation near the surface increased, and the amount of water migration inside the active layer decreased gradually. After the above four freeze-thaw stages were finished, the active layer completed a single freeze-thaw cycle. The main characteristics of soil temperature and moisture migration at different freeze-thaw stages are summarized in Table 2.

Table 2. Characteristics of the different freeze-thaw stages

| Stages | Definition | Initiation and termination | Soil temperature/moisture | Total number of measurements |
|---|---|---|---|---|
| ST | Summer thawing stage | Late April – early October (from when the active layer began to thaw downwards from the surface until the thawing process reached its maximum depth) | Soil temperatures in the active layer decreased from ground surface downwards; Moistures migrated downwards accompanied with the downward movement of the thawing front. | Eight soil depths with 60288 temperature data; seven soil depths with 52752 moisture data |
| AF | Autumn freezing stage | Initiated when the active layer reached its maximum thawing depth; Terminated when the whole active layer became frozen. | Temperatures of active layer were lower in its bottom or upper part and higher in its middle. Moisture in the thawed part of active layer migrated to both of the upper and lower freezing fronts and froze there. | Eight soil depths with 10752 temperature data; seven soil depths with 9408 moisture data |
| WC | Winter cooling stage | Initiated when the freezing process finished in late October; Terminated in the mid-late January of the next year. | Temperatures of active layer increased with the increasing depth. Moisture migration was not high due to low ground temperatures. | Eight soil depths with 35328 temperature data; seven soil depths with 30912 moisture data |
| SW | Spring warming stage | Initiated in early February; Terminated in late April. | Daily freezing and thawing cycles appeared on ground surface in late April. Ground temperature gradient decreased and the rate of unfrozen water | Eight soil depths with 34176 temperature data; seven soil depths with 29904 |

migration decreased gradually.     moisture data
Moisture content near the
ground surface showed a
decreasing trend.

Based on the observation data obtained from the experimental site in 2017 and 2018, the initiation and termination points, and the corresponding duration of each stage were calculated (Table 3). ST stage started on April 29, 2017, and ended on October 2, 2017, lasting 157 days. Meanwhile, AF stage was much shorter, lasting about 28 days. WC and SW had a similar duration of 92 and 89 days, respectively.

Table 3. The start-stop-time and duration of different freeze-thaw stages of the active layer

| Stage | start-stop time (yyyy/mm/dd) | time of length (days) |
|-------|------------------------------|-----------------------|
| ST | 2017/4/29-2017/10/2 | 157 |
| AF | 2017/10/3-2017/10/30 | 28 |
| WC | 2017/10/31-2018/1/30 | 92 |
| SW | 2018/1/31-2018/4/29 | 89 |

## 3.2 Dynamics of $R_s$ fluxes in different freeze-thaw stages of the active layer

At the Beiluhe experimental site, $R_s$ flux changed as the freeze-thaw processes of active layer developed, showing distinct dynamics in different freeze-thaw stages of the active layer (Fig. 3). $R_s$ flux showed a rapidly increasing trend as the thawing of the active layer intensified in the ST stage. The $R_s$ flux rate rose from 0.26 to 2.77μmol/m²s in 2017 and 0.53 to 2.82 μmol/m²s in 2018. In AF stage, $R_s$ flux fluctuated between 1.49 and 2.01μmol/m²s, although the number of observations was much smaller due to the short duration of the stage. In the following stage of WC, $R_s$ flux also decreased rapidly due to the prolonged lowered soil temperature, reaching the minimum values of 0.12 in 2017 and 0.13μmol/m²s in 2018 by the end of the stage. Then $R_s$ flux began to increase gradually as SW stage proceed. During SW stage, $R_s$ appeared as a small emission peak when the surface of the active layer underwent daily freezing and thawing cycles. Following the small emission peak, $R_s$ flux dropped a little and then started to ascend again quickly once ST arrived.

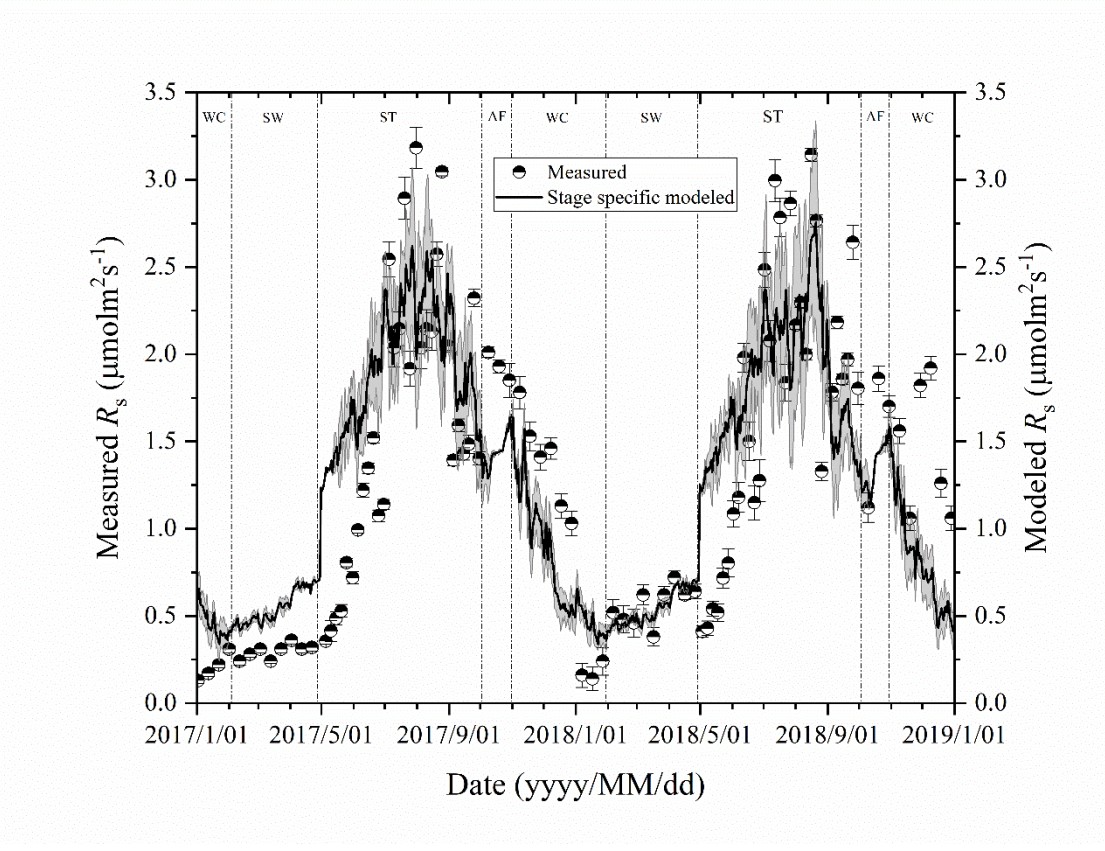

Fig.3. Variations of measured and modeled $R_s$ flux at different freeze-thaw stages in 2017 and 2018. Error bars show standard error of measured $R_s$ (n=6)

### 3.3 Contribution of $R_s$ in different freeze-thaw stages of the active layer

$R_s$ fluxes had significant correlations with soil temperature at 5 cm depth, and the exponential models explained the variations effectively except for AF stage (Fig. 4a). In contrast, soil water content was a poor predictor for $R_s$ with varying relationship for each freeze-thaw stage (Fig.4b). Meanwhile, daily average $R_s$ fluxes modeled by the exponential models for specific freeze-thaw stages were well matched with measured fluxes (Fig. 3). In addition, $RMSE$ analysis showed that the exponential models of soil respiration were preferable for $R_s$ prediction at different freeze-thaw stages ($RMSE<0.67$, Fig.5). As such, we calculated $R_s$ models, the temperature sensitivity ($Q_{10}$) and the sum of $R_s$ ($SR$) based on the Eqs. (1), (3) and (4) in four freeze-thaw stages during a complete freeze-thaw cycle from April 29, 2017 to April 28, 2018 (Table 4). The $SR$ emission during the ST stage ($1041.85\pm23.83$ $gCO_2/m^2$) was much higher than that during the other three stages ($150.54\pm6.80$ to $310.69 \pm12.33 gCO_2/m^2$). The relative contribution of $SR$ during each freeze-thaw stage to the total $R_s$ emission in a complete freeze-thaw cycle ($R_{cycle}$) ranged from $8.89\pm0.18\%$ to $61.32\pm0.32\%$. The $SR$ at AF stage was the lowest ($150.54\pm6.80$ $gCO_2/m^2$) and its contribution rate to $R_{cycle}$ was only $8.89\pm0.18\%$.

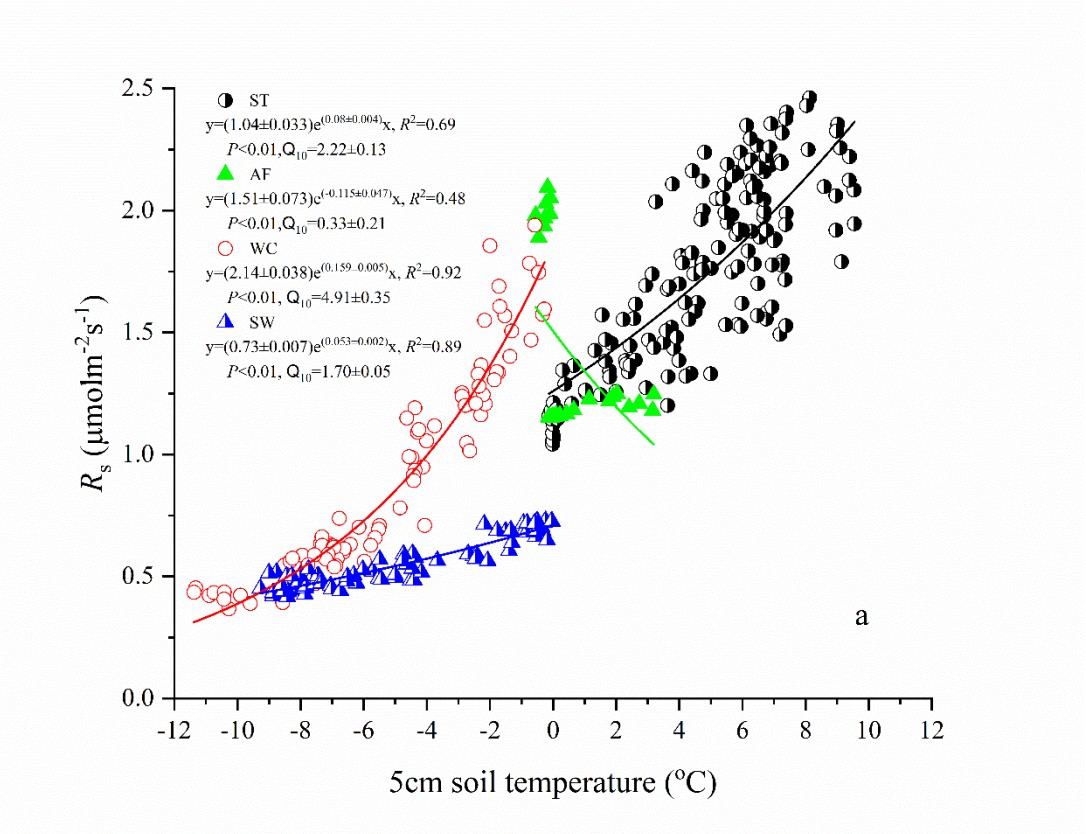

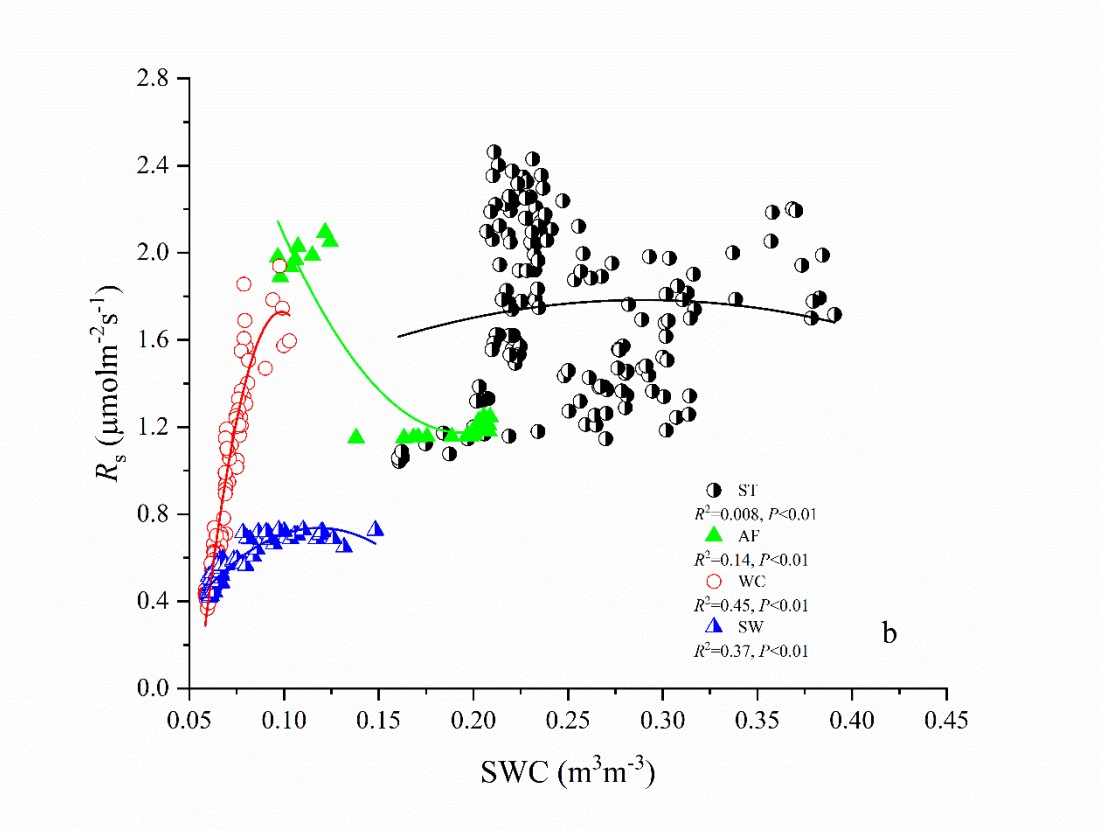

Fig. 4. Relationship between soil temperature (a) and moisture (b) at 5cm depth and measured $R_s$ flux for the summer thawing stage (ST), autumn freezing stage (AF), winter cooling stage (WC),

and spring warming stage (SW)

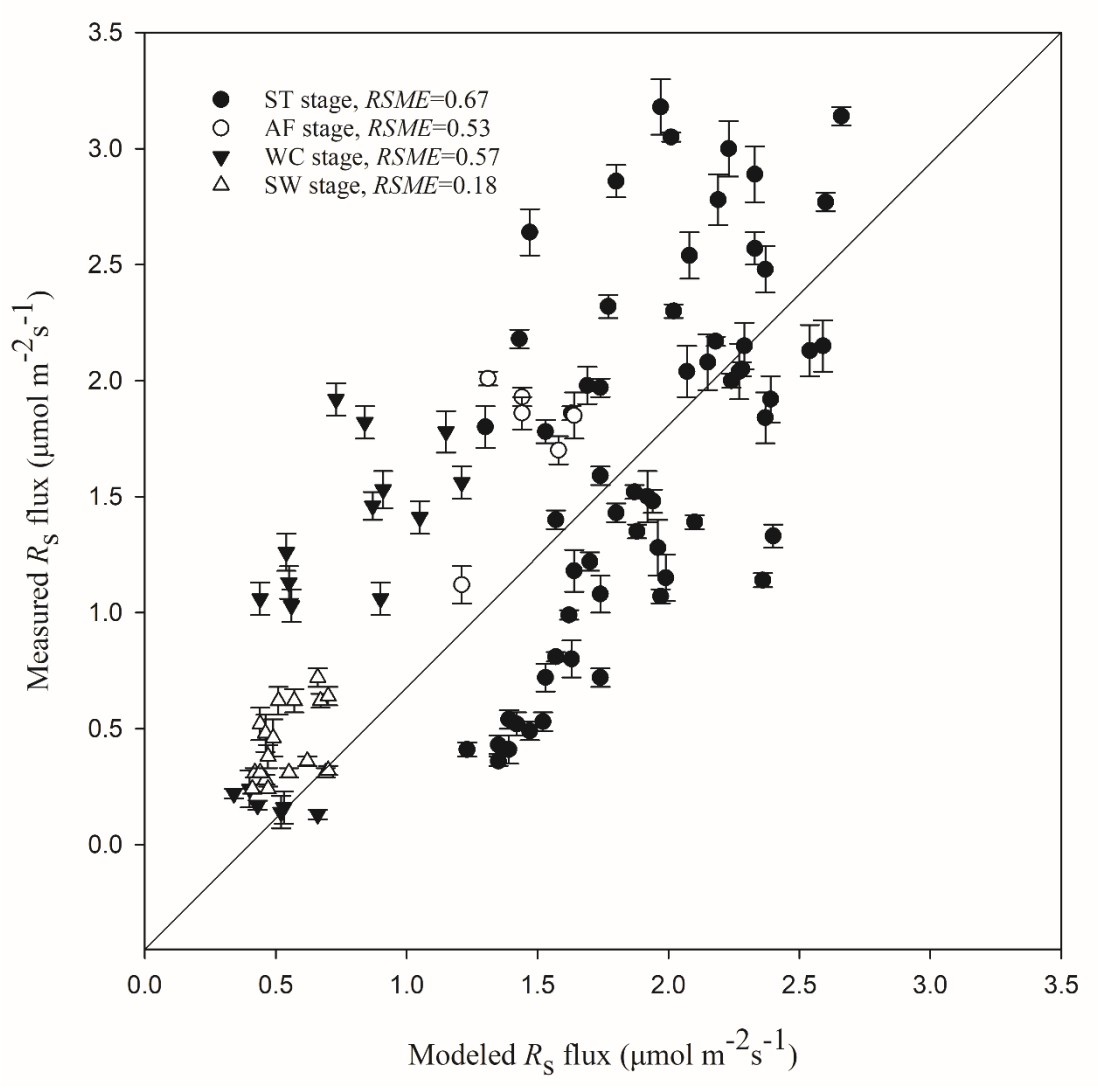

Fig.5. Modeled vs. measured $R_s$ fluxes at different freeze-thaw stages. Error bars represent
standard errors of measured $R_s$ flux (n=6). The solid line is a 1:1 line.

Table 4. The $R_s$ model, $Q_{10}$ value, $SR$ and its contribution to $R_{cycle}$ in different freeze-thaw stages

| Stages | $R_s$ model | | $Q_{10}$ | SR (gCO$_2$/m$^2$) | Rate of contribution to $R_{cycle}$ (%) |
|---|---|---|---|---|---|
| ST | $R_s = (1.04 \pm 0.033)e^{(0.08\pm0.004)T}$ | $R^2$=0.69 | 2.22±0.13 | 1041.85±23.83 | 61.32±0.32 |
| AF | $R_s = (1.51 \pm 0.073)e^{(-0.115\pm0.047)T}$ | $R^2$=0.48 | 0.33±0.21 | 150.54±6.80 | 8.89±0.18 |

| | | | | | |
|---|---|---|---|---|---|
| WC | $R_s = (2.14 \pm 0.038)e^{(0.159 \pm 0.005)T}$ | $R^2=0.92$ | 4.91±0.35 | 310.69±12.33 | 18.43±0.11 |
| SW | $R_s = (0.73 \pm 0.007)e^{(0.053 \pm 0.002)T}$ | $R^2=0.89$ | 1.70±0.05 | 189.90±8.47 | 11.29±0.11 |
| Total $R_s$ emission in a complete freeze-thaw cycle ($R_{cycle}$, gCO$_2$/m$^2$) | | | | 1692.98±51.43 | |

## 3.4 Factors affecting $R_s$ fluxes in different freeze-thaw stages

$R_s$ was positively correlated to soil temperatures, following an exponential relationship with the 5cm soil temperatures regardless of soil water status during the freeze-thaw stages. When calculated on the basis of the dataset of each stage, the $Q_{10}$ values were 2.22±0.13 ($R^2=0.69$), 0.33±0.21 ($R^2=0.48$), 4.91±0.35 ($R^2=0.92$), and 1.70±0.05 ($R^2=0.89$) for ST, AF, WC and SW with soil temperatures ranging from -0.13 to 9.55 °C, -0.55 to 3.19 °C, -11.38 to -0.28 °C, and -9.28 to -0.04 °C, respectively (Table 4). The variations of $R_s$ fluxes, determined by the exponential models we developed, exhibited different characteristics at each freeze-thaw stage during a complete freezing and thawing circle (Fig.6). In ST stage (Fig.6a), for example, the variations of $R_s$, $T_s$, and SWC were basically consistent. $R_s$ showed an increasing trend as $T_s$ and SWC at 5cm rose due to the active layer thawing from the surface, and reached the maximum (2.66±0.16 μmol/m$^2$s) until August. Then $R_s$ decreased with fluctuations as $T_s$ and SWC dropped. As AF began (Fig.6b), however, $R_s$ flux did not decrease any longer when it reached its lowest level (1.11±0.15 μmol/m$^2$s) even though $T_s$ and SWC dropped sharply in response to soil freezing. As freeze-thaw process developed, $R_s$ flux increased slightly and reached a relatively stable state (1.47±0.15 μmol/m$^2$s), although the $T_s$ and SWC continued to lower with fluctuations. When WC stage started (Fig.6c), $R_s$ decreased again with fluctuations as $T_s$ and SWC continuously decreased, although the active layer was completely frozen. At the end of WC, $R_s$ flux decreased to its lowest level (0.39±0.06 μmol/m$^2$s). In SW stage (Fig.6d), $R_s$ flux began to increase with fluctuations as the $T_s$ rose in response to soil warming, while SWC had no change as the surface soil still remained frozen in the earlier stage and fluctuated wildly due to daily freeze-thaw process in the later stage.

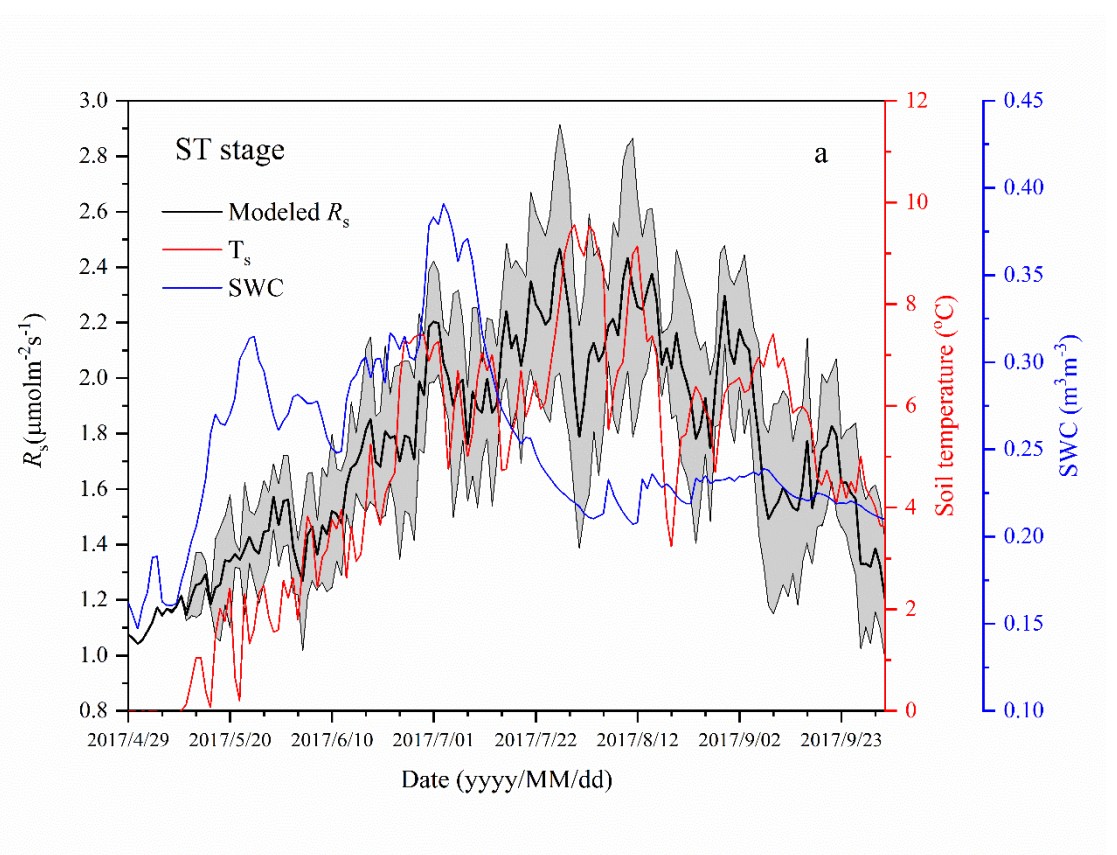

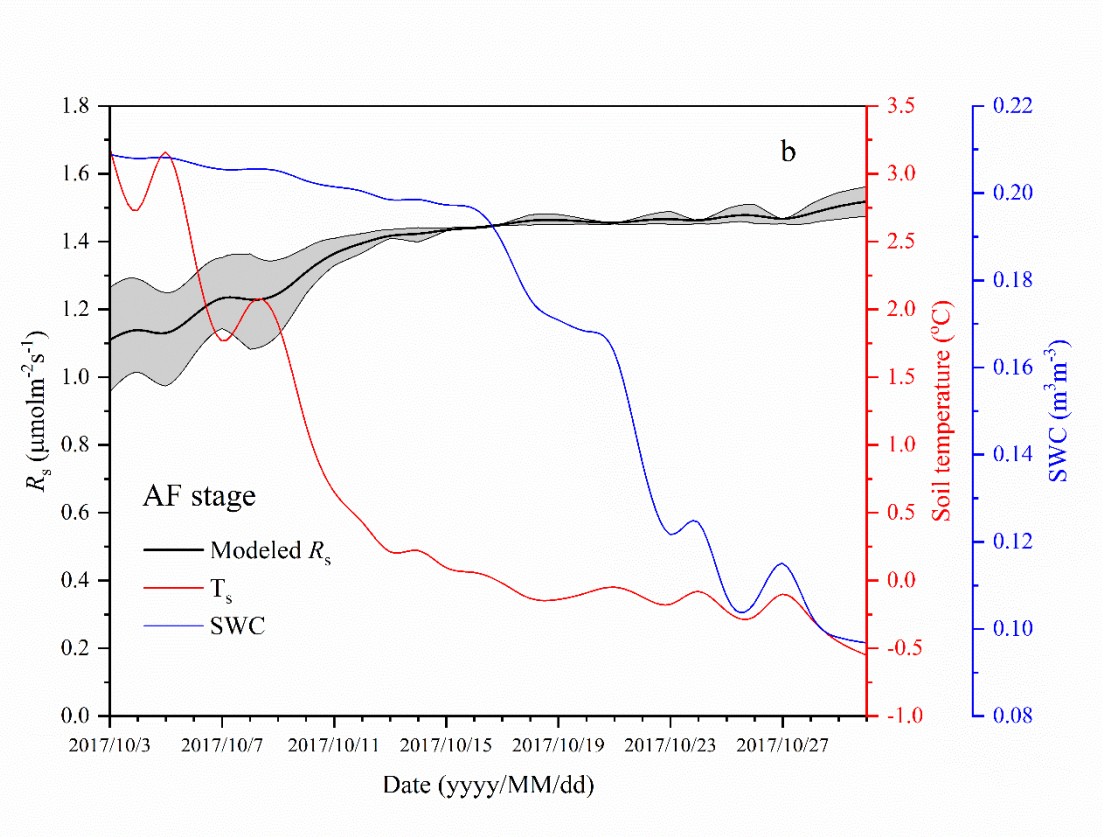

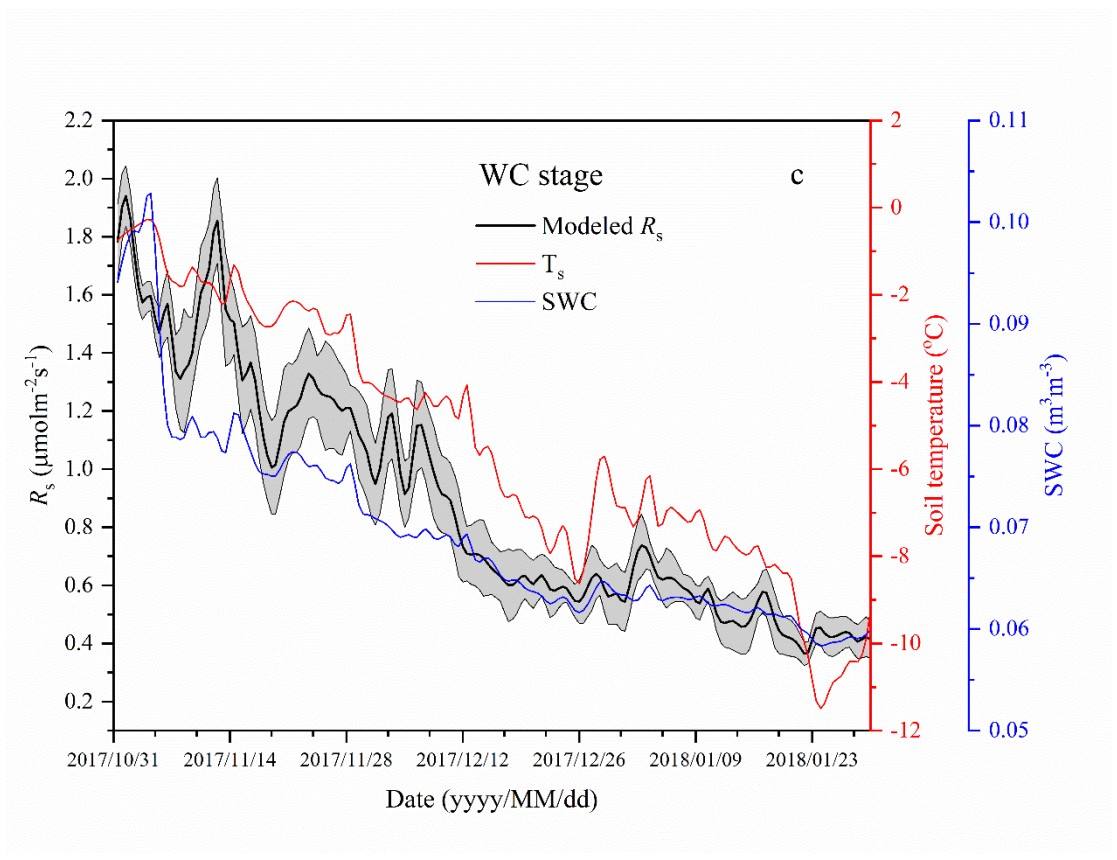

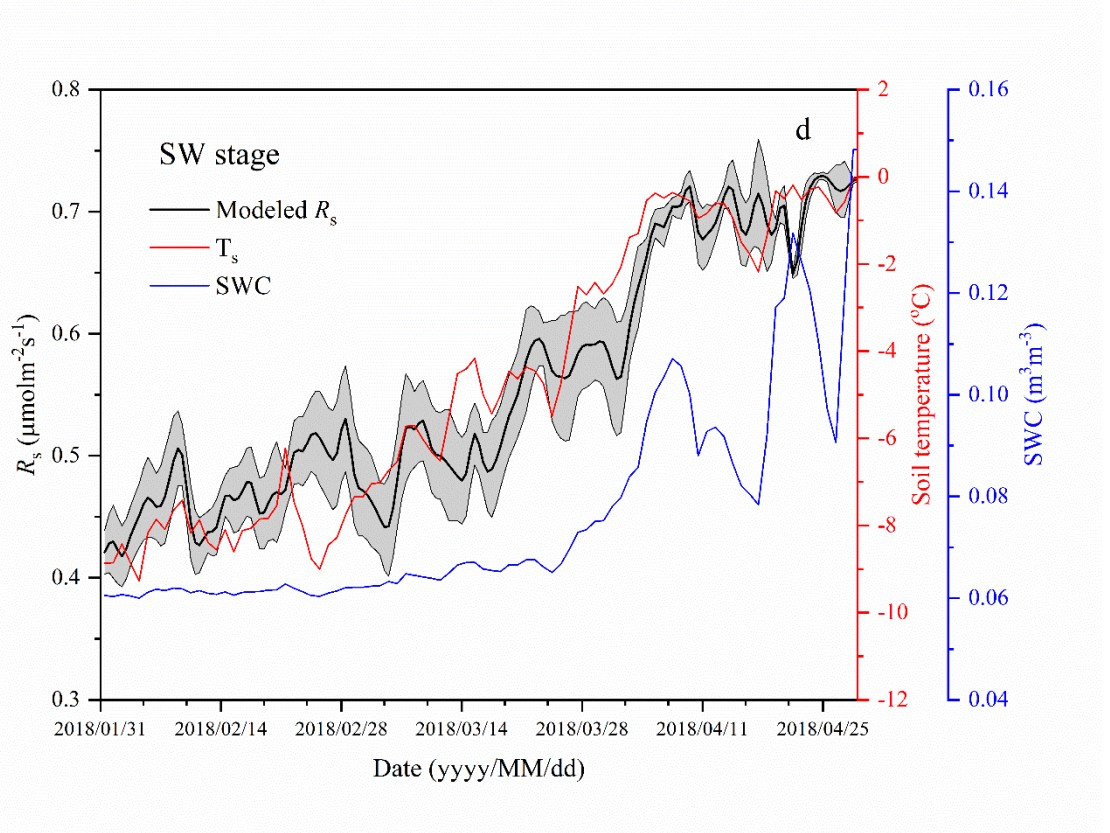

Fig.6. Variations in modeled soil respiration ($R_s$), soil temperature ($T_s$) and soil water content (SWC) for the four freeze-thaw stages including ST (a), AF (b), WC (c), and SW (d) in a

complete freeze-thaw cycle from later April 2017 to late April 2018. The SWC unit stands for water volume per total soil volume. The error band of modeled $R_s$ stands for 95% confidence interval.

## 4 Discussion

### 4.1 Impacts of freeze-thaw process on $R_s$ in different stages

Freeze-thaw process plays a significant role in soil biogeochemical processes in most high-latitude and high-altitude ecosystems (Grogan et al., 2004;Liu et al., 2016b). Furthermore, freeze-thaw effects on soil nutrient transformations may substantially influence the C balance of seasonally cold ecosystems (Grogan et al., 2004;Weih, 1998). Although many studies have shown that freeze-thaw events affect $R_s$ in tundra, boreal, and temperate soils (Liu et al., 2016a;Du et al., 2013), variations of $R_s$ in different freeze-thaw stages of active layer has not been studied in permafrost regions on the Qinghai-Tibet Plateau. Our observations clearly demonstrated that freeze-thaw process of active layer strongly affected the $R_s$ dynamics, and that $R_s$ emission models were significantly different for each of the freeze-thaw stages ($P<0.01$).

In ST stage, the active layer was mainly in a heat-absorbing and thawing condition, where heat was transferred from top to bottom and the thawing front also gradually migrated downwards (Zhao et al., 2000;Jiao and Li, 2014). We observed that $R_s$ was restrained by low temperature onset of the stage and then increased following an exponential correlation with soil thawing (Fig.6a and Table 4). Such increment of $CO_2$ fluxes after thawing was also observed in forests, alpine tundra, and arctic heath ecosystems (Wu et al.,2010a; Brooks et al., 1997; Elberling and Brandt, 2003). The rapid increase in $R_s$ flux may be due to the activation of root respiration and rhizospheric microbial respiration as well as heterotrophic respiration by enhanced microbial activities in bulk soil with greater C availability by expansion of thawing depth (Gaumont-Guay et al., 2006). In this stage, the dynamic changes in $R_s$ flux were not always associated with surface soil moisture content, which is reflected in a weak correlation between $R_s$ flux and moisture content ($R^2=0.008$, Fig.4b). Even $R_s$ flux reduced instead when soil moisture content increased quickly. The declining of $R_s$ flux at high soil moisture content conditions may be attributed to the increased free water in the soil clogging the soil pores as the thawed water accumulate at the thawing front (Zhao et al., 2000) and the rainy season of the Tibet Plateau (Ye, 1981). This hampers the two-way exchange of gases involved in respiration, with $CO_2$ passing upward from the soil to the atmosphere and oxygen moving in the opposite direction (Stepniewski et al., 1994). Although $R_s$ was hindered by a large increase in soil moisture content to some extent, more organic carbon and soluble matrix were exposed and became available to soil microbes as thawing deepened (Ping et al., 2008). As such, higher metabolic rates of soil organisms was stimulated and more soil carbon was released during this wet and humid ST stage (Keith et al., 1997). Furthermore, ten consecutive years of observation has found that the near-surface permafrost is warming in the alpine ecosystem regions underlain by permafrost on Qinghai-Tibet Plateau, increasing the duration of thawing by at least 14 days (Wu et al., 2015), which would emit significantly more $R_s$ in ST stage.

In AF stage, the $R_s$ flux showed a slightly increasing trend at initiation and came to a relatively stable state in the end, although the surface soil temperature declined sharply and the freezing process developed quickly (Fig.6b). When AF first began, the active layer was still basically an open system. It exchanged air and moisture with the atmosphere at least for some duration during the day, and the upper part of the active layer absorbed heat from the atmosphere during the daytime and

released it at night (Zhao et al., 2000). So although the reduction in soil temperature had some impacts on the soil microbial activities, some of the soil microbes were still active (Monson et al., 2006) and more nutrient matrix was decomposed and available due to diurnal freezing and thawing actions (Liu et al., 2016b). Furthermore, the freezing and thawing actions could initiate the activity of soil microorganisms and promoted the soil respiration (Grogan et al., 2004;Contosta et al., 2013). For these reasons, $R_s$ flux was maintained a relatively stable state accompanied by a slight increase at the initiation of AF. Although the soil of upper and bottom of active layer became frozen as the freezing process developed, the soil heterotrophic respiration rate was still higher in the thawing layer between the two freezing fronts, because the soil remained warm and unfrozen water was present (Olsson et al., 2003). As the two-way freezing process carried out and moisture migrated towards the freezing fronts (Zhao et al., 2000), the soil pores were constantly filled with ice, and as a result squeezed out and released the trapped $CO_2$ in the soil pores. Consequently, the $R_s$ still showed a relatively stable dynamic at the later period of AF.

In WC stage, once freezing process of active layer was completed, a sudden decline in soil temperature appeared along with a continuous decrease in soil water content (Fig. 2). The soil respiration at this cooling stage was influenced substantially because the continued decrease in soil temperature and liquid water content resulted in a partial death of microbes (Walker et al., 2006), lowered microbial activities and the reduction in substrate affinity (Nedwell, 1999). This appears to cause a continuous decline in the $R_s$ flux. However, we observed that the $R_s$ rate never reached zero although the $R_s$ flux continuously declined at this stage (Fig.6c), indicating that the soil microorganisms could maintain their activities at extremely low temperatures (Panikov et al., 2006), verifying the results of previous studies (Kurganova et al., 2007;Panikov and Dedysh, 2000). Furthermore, not all cells in the soil were killed or irreversibly damaged by the sustained low temperature and the dropping in soil water content (Walker et al., 2006), and the cold-adapted microflora still breathed and consumed the limited liberated nutrients in the frozen soil (Kurganova et al., 2007). Consequently, $R_s$ still maintained a detectable rate.

In SW stage, $R_s$ was still restrained by lower soil temperature and fluctuated at a low rate although the warming process of the active layer had begun as the air-temperature increased (Jiao and Li, 2014) before late March (Fig.6d). However, the rapid increase in $R_s$ flux started after late March, likely a result of the activation of soil microorganisms (Grogan et al., 2004) and the availability of nutrient matrix with increased soil water content (Liu et al., 2016b). This happened when the diurnal freezing-thawing process within the surface soil initiated and the surface soil water content increased due to thawing snow in this period (Zhao et al., 2000). This result was consistent with the reports on the Fenghuoshan region of the Qinghai-Tibet Plateau (Zhang et al., 2015a).

### 4.2 Dependence of $R_s$ on soil temperature in different freeze-thaw stages

Soil respiration is generally known to be controlled by soil temperature, soil moisture, or a combination of the two. When the soil moisture in not limited, soil respiration is mainly dependent on soil temperature (Zhang et al., 2015a). In the present study, the variations in soil temperature explained 48~92% of the freeze-thaw stage specific variabilities in $R_s$ fluxes. $Q_{10}$ values that reflect the quantitative relationship between $R_s$ and soil temperature differ at each freeze-thaw stage (Table 4). $Q_{10}$ values were higher at WC and SW stages when soil temperatures were lower than other stages (Fig.1) and at ST stage with higher soil moisture content (Fig.2). This result is consistent with a report from a temperate plantation forest where $Q_{10}$ values tended to be higher under lower temperature and higher soil moisture conditions (Yan et al., 2019). Negative correlations with soil

temperature and positive correlations with soil moisture of $Q_{10}$ values were also reported in a sub-alpine forest of the Eastern Qinghai-Tibet Plateau (Chen et al., 2010), and in a temperate cropland (He et al., 2016). However, $Q_{10}$ value was minimal and $R_s$ flux showed weak correlations both with soil temperature ($R^2$=0.48, $P$<0.01) and the soil moisture ($R^2$=0.14, $P$<0.01) at AF stage. This is mostly likely due to the fact that the active layer became an incomplete open system and hindered and even blocked the free exchanges of gas and moisture between the active layer and atmosphere during the later period of AF. Qinghai–Tibet Plateau is anticipated to be warmer and wetter under global warming (Li et al., 2010), accelerating permafrost thaw. This will enhance the temperature sensitivities of soil respiration at different freeze-thaw stages, resulting in much stronger response of the site to global warming in terms of $CO_2$ emissions.

As another influencing factor of soil respiration, soil moisture was reported to exhibit positive, negative or null effects on $R_s$ fluxes in various ecosystems (Gaumont-Guay et al., 2006;Balogh et al., 2011;Zhang et al., 2015a). In the present study, $R_s$ flux exhibited a low quadratic (positive) relationship with soil water content at the different freeze-thaw stages. Soil water content as an independent variable explained 0.8~45% of the variances of $R_s$ only (Fig.4b), suggesting that soil temperature was the main factor controlling variable for $R_s$ flux regardless of soil water status in all stages.

## 4.3 Cumulative $R_s$ in different stages and contributions to total C emission of a complete freeze-thaw cycle

The high determination coefficients of exponential equations for $R_s$ fluxes ($R^2$≥0.48) and the low $RMSE$ values ($RMSE$≤0.67) between the measured and modeled $R_s$ fluxes suggest that the freeze-thaw stage specific $R_s$ models proposed in this paper are accurate predictors of soil $CO_2$ emissions, at least in this region. The large amount of carbon emitted via soil respiration during the processes of freezing-thawing cycle of active layer suggests that $R_s$ of the Qinghai-Tibet alpine meadow ecosystem leads a significant carbon loss and may play an important role in global carbon cycle. According to the phenological changes of vegetation and the division of growing season and non-growing season of the Qinghai-Tibet Plateau (Xu et al., 2008a;Xu et al., 2005), heterotrophic respiration was most likely the main component of $R_s$ at AF, WC, and SW stages, which lie in non-growing seasons (from October to the next April); it accounted for almost two third of the ST stage, which belongs to a growing season (May to September). Because $Rs$ in a growing season differs substantially from those in non-growing seasons not only in quantity but also in key controlling variables, stage-specific carbon processes should be taken into consideration for accurate estimation of carbon sink/source of the Qinghai-Tibetan alpine ecosystem.

At ST stage, the modeled $R_s$ fluxes ranged from 1.23 to 2.66 μmol/m$^2$s, and the cumulative $R_s$ (1018.02–1068.68 gCO$_2$/m$^2$) was approximately estimated to be 61% of the total $R_s$ emission in a complete freeze-thaw cycle (1641.55 to 1744.41 gCO$_2$/m$^2$). The cumulative $R_s$ of the ST stage and its contribution to the total $R_s$ emission in a complete freeze-thaw cycle in our study were both higher than those at the Fenghuoshan region on the Qinghai-Tibet Plateau or those from Arctic tundra (Zhang et al., 2015a;Elberling, 2007). This is likely due to the longer duration (157 days) of ST stage at our study site and the unique seasonal climate of the plateau. More specifically, the Tibetan alpine meadow receives more than 60–90% of the total precipitation during ST stage, with less than 10% in the other stages (Xu et al., 2008b). In addition, higher soil temperature and water content in wet and humid summers stimulate microbial activity, inducing higher metabolic rates of soil

organisms and roots (Keith et al., 1997). Furthermore, observation made in ten consecutive years in the alpine ecosystem regions underlain by permafrost on the Qinghai-Tibet Plateau has found that warming of the near-surface permafrost increased the duration of thawing by at least 14 days (Wu et al., 2015), emitting significantly greater amount of $R_s$ in ST stage.

At AF stage, exponential regression analysis was carried out with fewer measured $R_s$ values because the duration was shorter than other stages. The modeled $R_s$ fluxes were generally lower than that measured during this stage (Fig.3). These biases between the measured and modeled $R_s$ fluxes were likely to be caused by sampling scheme. The low sampling frequency (two occasions for daily average data) in the period of AF could increase the variance of aggregated estimates (Ryan and Law, 2005). In addition, measurements during this stage were usually restricted to daytime and dry days, and the sampling would inevitably miss the pulse of microbial or root activity immediately following occasional precipitation (Doff sotta et al., 2004). Thus, the cumulative $R_s$ (143.74 to 157.34 $gCO_2/m^2$) calculated by an exponential model only accounted for about 8.89% of the total $R_s$ emission in a complete freeze-thaw cycle, which probably underestimated the $R_s$ emission during this stage. Although the active layer gradually became a closed system in this stage, it is noteworthy that a proportion of respired soil $CO_2$ can still be transported via vascular plants, which may function as a conduit for $CO_2$ from deeper soil layers (Ström et al., 2005). Furthermore, the diurnal freezing and thawing actions occurring in this stage also played an important role on the $R_s$ emissions (Contosta et al., 2013). Therefore, more frequent observations with automated chambers incorporating vegetation function are warranted to refine the estimated $R_s$ at AF stage in this study.

At WC stage, $R_s$ fluxes continuously descended with fluctuation from 1.92 to 0.13 $\mu mol/m^2s$ and the cumulative $R_s$ (298.36–323.02 $gCO_2/m^2$) was estimated to be 18.43±0.11% of the total $R_s$ emission in a complete freeze-thaw cycle. The continued decrease in soil temperature and liquid water content appeared to lead the decline in soil respiration. However, little is known regarding the root function of the alpine meadow and soil microbial activities at this stage. It is generally assumed that lowering soil temperature may hinder microbial activities, but laboratory-based experiments have found that microbial activities can still be substantial at low temperatures of -6°C to -10 °C (Panikov and Dedysh, 2000;Walker et al., 2006). Further research on the physiology and roles of such psychrophilic microorganisms in soil respiration at WC stage is necessary.

At SW stage, the modeled $R_s$ fluxes showed a rising trend in the ranges from 0.42 to 0.72 $\mu mol/m^2s$, and were generally higher than those measured in 2017 (Fig.3). These biases may also be caused by low sampling frequency and simple averaging for daily average data in regression analysis (Ryan and Law, 2005). The diurnal freezing-thawing process during this stage also stimulated the activities of soil microorganisms and promoted the $R_s$ emissions, but the low sampling frequency and the restricted sampling time (between 9:00 and 11:30 a.m. local time) probably missed the peaks and pulses of $R_s$ fluxes of a day. Therefore, the measured $R_s$ flux may underestimate the actual emission rate in this stage. However, it is also reported that biases of chamber-based estimates of $R_s$ can be reduced by using a regression model which is extrapolated with soil temperature and moisture (Ryan and Law, 2005). In addition, the smaller value of *RMSE* at this stage also testified the temperature-driven model was preferable for the $R_s$ prediction (Fig.5). Thus, the cumulative $R_s$ (181.43–198.37 $gCO_2/m^2$) calculated by the exponential model was estimated to be 11.29±0.11% of the total $R_s$ emission in a complete freeze-thaw cycle. Despite of the possible biases between the modeled and measured $R_s$ fluxes, our model is still a reliable estimate for $R_s$ emission during this stage, which is further supported by the same trends in the variations between

the two values. The increasing trend in $R_s$ fluxes can be caused by the following mechanisms: First, the activation of soil respiration was mediated by increased soil microbial activities as soil temperature and water content increased. Furthermore, as spring proceeds with warming of soil, the mobilization of stored carbohydrates enhanced soil respiration (Davidson et al., 2006). Finally, daily freeze-thaw actions in late April may have further enhanced the soil respiration quickly.

## 5 Conclusions

The freezing and thawing process of active layer significantly controlled the soil respiration of the alpine meadow in permafrost region of the Qinghai-Tibet Plateau. The soil temperature was the key factor affecting soil respiration regardless of soil water status during each freeze-thaw stage. The cumulated soil respiration in different freeze-thaw stages ranged from 150.54 to 1041.85 $gCO_2m^{-2}$, and the cumulated soil respiration in ST, AF, WC, and SW stages contributed about 61.32, 8.89, 18.43, and 11.29% to the total $R_s$ emissions in a complete freeze-thaw cycle, respectively. The $Q_{10}$ values were higher at WC and SW stages with lower soil temperatures, and at ST stage with higher soil moisture content. As the Qinghai–Tibet Plateau becomes warmer and wetter (Li et al., 2010), soil respiration at different freeze-thaw stages are predicted to be more sensitive to temperature. Furthermore, in the future climate of warmer temperatures, great changes in freeze-thaw process patterns may have important impacts on $R_s$. Further research is required to define the regulatory mechanism and its key processes on $R_s$ in different freeze-thaw stages of the active layer. In addition, due to short duration of the AF, more frequent observations should be carried out in order to more accurately evaluate the contribution of $R_s$ at the stage.

## Acknowledgments

This study was supported by the National Natural Science Foundation of China (41771080, 41701066, 41003032), the Fund of State Key Laboratory of Frozen Soil Engineering (No. SKLFSE-ZT-36) and the grant of China Scholarship Council. We are grateful to Dr. Yali Liu and Yu Gao for their help in the field measurement of soil $CO_2$ flux. We gratefully thank the reviewers for their comments. We also thank Jiyoung Kang for English correction.

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
