# Peer review of "Soil respiration of alpine meadow is controlled by freeze-thaw processes of active layer in the permafrost region of the Qinghai-Tibet Plateau"

_The Cryosphere, 2019_

## Referee Comment (RC1) · Anonymous Referee #1 · 12 Nov 2019

Wang et al. investigated the influence of freeze-thaw processes on seasonal $CO_2$ respiratory fluxes from a permafrost-affected ecosystems on the Qinghai-Tibet Plateau (QTP). Winter $CO_2$ fluxes, the effect of freeze-thaw processes as well as the QTP are understudied. The authors found that different freezing stages affect soil respiration dynamics differently but that soil temperature is the most important driver independent of soil water status. The manuscript therefore addresses relevant scientific questions within the scope of the journal. Impressive are the regular flux measurements during the frozen winter period. The manuscript, however, needs revision with regards to the presentation of figures and discussion of the results.

Tables 1 and 2 were missing from the manuscript. But from the text I understand that some of my comments below may have already been addressed in the tables.

**Major comments**

P1L21-23: The abstract contains quite a few abbreviations which make it somewhat difficult to quickly grasp the major points of the manuscript. None of the different freeze-thaw stages are defined nor the abbreviations explained.

P4L143-150: Some more information about chamber volume, chamber closing time, and flux calculations might be beneficial. What was the minimum flux the chamber system was able to detect? Did you have to increase the chamber closing time in winter in order to achieve the needed minimum change in $CO_2$ concentration for flux calculations? Do you have any concerns about the disturbance caused by collar installation and above-ground plant removal for the determination of soil respiration? What about the roots left in the soil after plant removal? It has been shown that roots respiration as well as the decomposition of dying/dead roots can affect soil respiration for some time after plant removal (e.g. Subke et al., 2006).

P9: During the ZC sub-stage, several possible scenarios are offered for the sudden increase in Rs. How many in situ measurement points actually fall within this very short period? From Figure 2 it looks like only one, maybe two points. How does this affect uncertainty for the developed Rs model? Did you also measure a diurnal freeze-thaw pattern during this period? How long after thaw during the day did you do the Rs measurement? Could substrate availability and soil aggregation be affected by freeze-thaw cycles and thus affect microbial activity during this stage?

Figure 4: Are these results based on the Rs model? Can you give an uncertainty range? On how may measurement points is the AF stage based? On the y-axis; should it read Rs and $\mu$mol? Is SWC the same as volumetric water content, i.e. ratio of water volume to soil volume? What is the dotted line in the AF subplot?

Using the different freeze stages and the developed Rs model in combination with the observed changes in active layer dynamics described by Wu et al. (2015) for the site, can you estimate or comment on by how much Rs has changed (or will change) and what period/freeze stage the changes are largest?

**Minor comments**

The usage of units differs between the text and figures. The date format between Figure 1, 2, and 4 is inconsistent.

P1L39: I suggest to start the sentence with "Furthermore, […]"

P2L47: […] on the northern hemisphere […]

P2L47-50: The numbers for the circumpolar SOC stocks mentioned here seem to be the numbers from Tarnocai et al. (2009)? Then they should be referenced. You could also consider updating the numbers published by Hugelius et al. (2014).

P2L50: Do you mean "sensitivity"?

P2L57: What do you mean with "completed"?

P3L91-93: If this is the hypothesis of this manuscript, I suggest using "should" instead of "must"

P3L106: Is all that precipitation falling as rain? Is the study site covered in snow in the winter?

P3L116: How large is the chosen terrain? Does the active layer observation site contain one or multiple locations with soil temperature and moisture probes?

P4L147: Did you use an automated system or manual measurement?. Nevertheless, a measuring interval of 3-7 days is still quite impressive, especially during the frozen periods.

P4L156-157: Was SWC determined at every measurement day, including when the soil and/or the soil water was frozen? Does the soil moisture probe detect only liquid water? Is SWC based on soil or on pore volume?

P5L179: Do you present and/or discuss the ANOVA results in the manuscript?

Section 3.1. A summary table with an explicit definition of the different stages, a description of the characteristic temperature and soil moisture profiles as well as the timing and length of each freezing stage might be beneficial. You could also include the number of measuring points for each period.

P6L28: What do you mean with "regularly"?

Section 3.3: Table 2 was missing from the manuscript. However, a summary table of the Q10, mean (min/max) Rs rates, SR, and contribution to the annual balance would be helpful. Then the numbers do not necessarily have to be mentioned in the text, which could help with readability. Figure 3 could then possibly be presented in Table 2 as well? For Q10, it would also be important to indicate the range of soil temperatures during the different stages, since the effect of low temperature on temperature sensitivity and Q10 is later discussed in the text.

P7L274: "shown"

P7L286-287: Did you test the effect of soil moisture on Rs? In the manuscript you state that Ts is the most important factor.

P8L312: Is the exponential increase in abundance of microbes, which are adapted to freezing conditions, with increasing temperature true for all freezing stages? Would that mean that these microbes also dominate during the summer?

P8L320: I would assume that a higher Q10 would then specifically matter in winter if the winter was warming more than the summer?

P8L325: What was the limit for soil water content to be "sufficient"?

P14L11: "[…] late April 2017 […]"

Reference list need checking. Some reference are not edited correctly (e.g. Grogan, P., and Chapin III, 2000)

Figure 1: It could be more intuitive to reverse the y-axis to indicate soil depth. In addition, it would be helpful to also indicate the timing of the different stages along the x-axis (similar to Figure 2)

**References**

Hugelius, Gustaf, et al. "Estimated stocks of circumpolar permafrost carbon with quantified uncertainty ranges and identified data gaps." *Biogeosciences (Online)* 11.23 (2014).

Subke, Jens-Arne, Ilaria Inglima, and M. Francesca Cotrufo. "Trends and methodological impacts in soil CO2 efflux partitioning: a metaanalytical review." *Global Change Biology* 12.6 (2006): 921-943.

Tarnocai, Charles, et al. "Soil organic carbon pools in the northern circumpolar permafrost region." *Global biogeochemical cycles* 23.2 (2009).

Wu, Qingbai, et al. "Changes in active-layer thickness and near-surface permafrost between 2002 and 2012 in alpine ecosystems, Qinghai–Xizang (Tibet) Plateau, China." *Global and Planetary Change* 124 (2015): 149-155.

---

## Referee Comment (RC2) · Anonymous Referee #2 · 19 Nov 2019

Junfeng Wang et al. investigated soil respiration in an alpine meadow of the Qinghai-Tibet Plateau (QTP) over almost two complete years. This investigation produced an impressive data-set of CO2 soil respiration fluxes, which is, especially for winter-time, unique for this region and of high importance for permafrost regions in general. This data-set was used to study the impact of freeze-thaw processes on the soil respiration fluxes and the different fluxes during different freeze-thaw stages are shown. Furthermore, the regulation of soil respiration by other parameters is shown. To address the relevant scientific questions within the scope of the journal, the manuscript needs major revision with regards to description of the methods, the discussion of the results and maybe to the usage of this impressive data-set.

**Major comments**

In general, the title seems to be a bit misleading as the impact of freeze-thaw processes on soil respiration fluxes is not as obvious to me in the manuscript as stated in the title. Of course, it makes sense to partition the year-round measurements into different freeze-thaw stages (and for sure, there are flux differences between the freeze and thaw stages) and also it's worth to discuss the role of freeze-thaw processes on the fluxes. However, the main driver of these fluxes is still the soil temperature, which is already widely known. If the authors want to point out the significant role of freeze-thaw processes, they should state this more clear throughout the manuscript and bring more (statistical) evidence of it's impact on the fluxes. So far, the authors have shown an increase of soil respiration in the ZC substage, which might be attributed to the freezing process. However, the amount of outgassed CO2 during this period make up much less than 5% to the annual budget and the data points during this period seem to be really sparse. During the WC, SW and ST stage the freezing and thawing processes seem to be of minor importance to the soil respiration, even though an impact during the SW stage is discussed but evidence for this impact is missing in the manuscript. Therefore, the authors might shift the focus of the manuscript towards a model-based budget of soil respiration on an annual basis (see next paragraph) or they bring evidence on a statistical basis on the regulation of freeze and thaw processes on Rs fluxes.

The flux data-set is impressive, especially as it is really difficult to conduct these chamber-based measurements during winter-time and regularly over a two-year period. Can the authors state something about similar data-sets in such areas (alpine, permafrost-affected)? Especially the winter-time soil respiration fluxes would be of interest to the reader here. How high/low are these fluxes compared to other regions and why? Aren't there other soil respiration fluxes from such areas from this pedon-scale? Then the authors should point out this uniqueness of the data-set as the winter-time Rs fluxes make up about 30% of the annual fluxes. A modeling of the Rs fluxes has been done, but from the text it remains unclear which model is used to calculate the budget (model from equation 1 or interpolation of average Rs flux rate(described at line 170)) and where the (modeled?) fluxes shown in figure 4 come from. However, to calculate an annual budget, an interpolation of average Rs fluxes seems to be not sufficient, while temperature-based respiration models are widely used to calculate flux budgets. Furthermore, the interannual-variability of the Rs fluxes between the two years might be worth to look at. Are there differences in the budgets and if so, why (e.g. it seems like the Rs fluxes from the SW stage are significantly higher in the second year)?

Some more information on soil and vegetation composition of the chamber set-up would be helpful to the reader (especially when the fluxes are compared to those from other regions). What soils are generally found in this area? Are they organic-rich/poor? What is the active layer depth? How deep are the main rooting zones of the vascular plants? If the roots mainly reach e.g. about 20cm into the soil, the insertion depth of the PVC collar might be too low as lateral roots still reach into the chamber collar and may alter the measured respiration flux. Furthermore, the closure time of the chamber is of interest. Where they similar during winter and summer-time? If the plants inside the collars were removed just one day before the measurements started, there might be some artefacts due to this disturbance (Diaz-Pines et al., 2010) that need to be taken into account. In general, a critical review of the clipping method should get more attention and it should be stated why this method was applied instead of other less disturbing methods (Subke et al., 2006). Furthermore, the reader needs to know something about the flux calculation procedure? Was a linear or an exponential model used to calculate the fluxes? Based on which quality criteria (check Görres et al., 2014)?

Two tables are missing in the manuscript. As they seem to contain a lot of information on flux details, they may already answer some of the question that are stated in this review.

**Minor comments**

In the abstract some abbreviations are used without an introduction, which needs to be changed.
If the authors shift the focus of the manuscript, the abstract should be changed accordingly.

Line 40: At least one citation is needed here.

Line 114: To compare the fluxes from this region with other regions it would be good to say something about the soils (carbon contents, C/N, etc) beside a detailed vegetation description.

Line 119: What about the soil moisture probes at different depths? Why were they inserted as in the end just the SWC at 5cm was used?

Line 144: Are there no differences in vegetation cover, soils, etc. so that one measurement plot in the six 5x5m measurement plots can serve as replicates? If not, there might be a chance of discussing other impacts such as carbon content, vegetation cover and more on the Rs fluxes. Anyway, a detailed description of soils and vegetation is needed here.

Line 148: What have the authors done with re-growth of plants during the measurement period. For sure, there have been some.

Line 159: Unfortunately, it remains unclear which model was used for calculating the contributions of Rs from each freeze-thaw stage to the annual budget. This must be stated clearly. So far it reads, that the resulting fluxes from equ.1 were used to describe the dependency of Rs on T, while for the budget calculation interpolated average fluxes were used. If a model exist, why interpolated averages were used then? May it would make more sense to use a temperature-based model and, as Q10 was also used in the manuscript and it is shown that there are differences between the different stages, to also include Q10 into a model (e.g. Eckhardt et al., 2019).

Line 179: ANOVA is described here but not referred to later in the text.

Line 228: Yes, there are freezing and thawing processes in the active layer, but the suggestion that they strongly regulate the Rs fluxes seem to be a bit speculative as the authors don't bring any evidence here (again, some statistics would be helpful), that there is a regulation of Rs fluxes by these processes (and should therefore be part of the discussion and not of the results). The only argument is that the freeze-thaw processes are taking place at the same time when the Rs fluxes are starting to rise (which might be simply due to rising temperature).

Line 308: Can the autotrophic respiration act as reason for the differences in Q10 here? Due to the clipping of the vegetation in the chamber plots, there shouldn't be any, right?

Line 375: As there is no clear evidence for a regulation of the Rs fluxes, the authors should be more carefully use the term 'significantly' to describe this relationship (or refer to ANOVA?). For sure, there are significant differences between the Rs fluxes from the different freeze and thaw stages, but are they really driven by the actual freezing and thawing processes or just driven by different soil temperatures of the stages?

Figure 1: Additionally, the authors should include the freeze and thaw stages in the graph

Figure 2: The authors should use a consistent date string (compared to figure 1). Furthermore, drawed lines in the graph would give a better readability to see which Rs fluxes belong to which stage.

Figure 3: Which year are those flux contributions from? Why not for both years? May a mean value would be better practice?

Figure 4: From which model are these Rs fluxes shown here? Are the SWC values relevant (if so, why aren't they included in a model?; if not, why are they shown?)?

Literature

Diaz-Pines, E., Schindlbacher, A., Pfeffer, M., Jandl, R., Zechmeister-Boltenstern, S., and Rubio, A.: Root trenching: a useful tool to estimate autotrophic soil respiration? A case study in an Austrian mountain forest, Eur. J. For. Res., 129, 101–109, https://doi.org/10.1007/s10342-008-0250-6, 2010.

Subke, J.-A., Inglima, I., and Cotrufo, M. F.: Trends and methodological impacts in soil CO2 efflux partitioning: A metaanalytical review, Glob. Change Biol., 12, 921–943, https://doi.org/10.1111/j.1365-2486.2006.01117.x, 2006.

Görres, C. M., Kutzbach, L., and Elsgaard, L.: Comparative modeling of annual CO2 flux of temperate peat soils under permanent grassland management, Agr. Ecosyst. Environ., 186, 64–76, https://doi.org/10.1016/j.agee.2014.01.014, 2014.

Eckhardt, T., Knoblauch, C., Kutzbach, L., Holl, D., Simpson, G., Abakumov, E., and Pfeiffer, E.-M.: Partitioning net ecosystem exchange of CO2 on the pedon scale in the Lena River Delta, Siberia, Biogeosciences, 16, 1543–1562, https://doi.org/10.5194/bg-16-1543-2019, 2019.

---

## Author Comment (AC1) · 30 Dec 2019

**Responses to reviewer's comment**

We are grateful to anonymous reviewers for their helpful and constructive suggestions and comments. We have tried our best to address all issues raised by them, which we believe improve the manuscript substantially. The revised contents and detailed responses to the valuable questions are as follows:

**Reviewer 1 stated that;**

P1L21-23: The abstract contains quite a few abbreviations which make it somewhat difficult to quickly grasp the major points of the manuscript. None of the different freeze-thaw stages are defined nor the abbreviations explained.

**The reviewer made a valid point, and we spelled out the words and the corresponding abbreviations of the different freeze-thaw stages. The revised abstract is as follows:**
**"Freezing and thawing action of the active layer plays a significant role in soil respiration ($R_s$) in permafrost regions. However, little is known about how the freeze-thaw process regulates the $R_s$ dynamics in different stages for the alpine meadow underlain by permafrost on the Qinghai-Tibet Plateau (QTP). We conducted continuous in-situ measurements of $R_s$ and freeze-thaw process of the active layer at an alpine meadow site in the Beiluhe permafrost region of QTP to determine the regulatory mechanisms of the different freeze-thaw stages of the active layer on the $R_s$. We found that the freezing and thawing process of active layer modified the $R_s$ dynamics differently in different freeze-thaw stages. The mean $R_s$ ranged from 0.56 to 1.75μmol/m²s across the stages, with the lowest value in the spring warming (SW) stage and highest value in the summer thawing stage (ST); and $Q_{10}$ among the different freeze-thaw stages changed greatly, with maximum (4.9) in the winter cooling stage (WC) and minimum (1.7) in the SW stage. Patterns of $R_s$ among the ST, autumn freezing (AF), WC, and SW stages differed, and the corresponding contribution percentages of cumulative $R_s$ to annual total $R_s$ were 61.54, 8.89, 18.35, and 11.2%, respectively. Soil temperature ($T_s$) was the most important driver of $R_s$ regardless of soil water status in all stages. Our results suggest that as the climate warming and permafrost degradation continue, great changes in freeze-thaw process patterns may trigger more $R_s$ emissions from this ecosystem because of prolonged ST stage."**

P4L143-150: Some more information about chamber volume, chamber closing time, and flux calculations might be beneficial. What was the minimum flux the chamber system was able to detect? Did you have to increase the chamber closing time in winter in order to achieve the needed minimum change in $CO_2$ concentration for flux calculations? Do you have any concerns about the disturbance caused by collar installation and above-ground plant removal for the determination of soil respiration? What about the roots left in the soil after plant removal? It has been shown that roots respiration as well as the decomposition of dying/dead roots can affect soil respiration for some time after plant removal (e.g. Subke et al., 2006).

**To clarify them, we added additional explanations and elaborate the method section. In our**

experiment, we used the LI-8100A Automated Soil Gas Flux System to determine the soil respiration ($CO_2$ flux). The smart chamber is a portable, self-powered 20 cm survey chamber featuring an embedded microprocessor and internal storage for real-time flux calculations when configured with LI-COR gas analyzers. The LI-8100A is a very popular survey measurement device for $CO_2$, $CH_4$, $N_2O$ and other trace gases depending on the trace gas analyzer installed. In our experiment, the LI-8100A was only assembled with a $CO_2$ gas analyzer. The measuring range for $CO_2$ that the infrared gas analyzer can detect is from 0 to 20000μmol/mol with accuracy 1.5% of reading.

Throughout the measurements, we adopted the recommended settings by the LI-COR to determine the soil respiration flux. A typical measurement protocol was applied: Obs. Length: 2 mins, Dead band: 25 seconds, Pre-purge: 30 seconds, Post-purge: 45 seconds, Chamber Volume: automated, IRGA volume/total volume: automated. Chamber offset of the program was adjusted to 2 cm.

During the experiment, we carefully designed the experiment and gave full consideration to the potential disturbance to soil respiration caused by collar installation and plant removal. In order to minimize the disturbance for the determination of soil respiration, we installed the collars one month before the experiment and left all the collars permanently inserted into the soil. In addition, after the above-plant was clipped and left undisturbed for more than 24 hours before we initiate the measurement for soil respiration. This "resting time" allowed the removal of any excess $CO_2$ released by roots disturbed during above-plant removed.

Soil respiration includes $CO_2$ produced from processes such as root expansion, mycorrhizal exploration, and microbial decomposition of litter and soil organic matter (Phillips and Nickerson, 2015). We did consider the paper entitled with "Trends and methodological impacts in soil $CO_2$ efflux partitioning: A meta-analytical review" by Subke et al. (2006) that was appeared in Global Change Biology. They summarized impacts of different methodologies on soil $CO_2$ efflux partitioning and found there was no coincident influences of plant removal on soil respiration. However, Guo et al. (2011) found that mowing of meadow would increase soil respiration (appeared in Journal of Acta Agrestia Sinica). Thus, the above-plant removal in our experiment could have increased soil respiration and our results may have overestimated soil respiration due to disturbance of the plots. In order to minimize the disturbance to soil respiration, the living plants inside the collar were carefully removed at the soil surface at least 1 day prior to the measurement.

In addition, we supplemented detailed information of the measurement protocol for determining soil respiration using LI-8100A device in the manuscript.

P9: During the ZC sub-stage, several possible scenarios are offered for the sudden increase in Rs. How many in situ measurement points actually fall within this very short period? From Figure 2 it looks like only one, maybe two points. How does this affect uncertainty for the developed Rs model? Did you also measure a diurnal freeze-thaw pattern during this period? How long after thaw during the day did you do the Rs measurement? Could substrate availability and soil aggregation be affected by freeze-thaw cycles and thus affect microbial activity during this stage?

We would like to point out that each open circle in Figure 2 represents the average $R_s$ obtained

from six plots on each sampling day. The daily average $R_s$ flux between the sampling dates was obtained by interpolating the average $R_s$ flux rate. Using the observed data and interpolated data, coupled with soil temperatures at different layers recorded every 30 min by a data logger, we developed the $R_s$ model at different freeze-thaw stages. We ran regression analysis, and found that the value of adjusted $R$ was high enough that the $R_s$ model was accurate and reliable at 95% confidence level.

At our study site, we used a data-logger powered by solar panel to automatically record the soil temperatures at different layers every 30 min. The depths for the soil temperature measurement were 5, 20, 50, 80, 120, 150, 180 and 230 cm. We believe that the diurnal freeze-thaw changes were clearly detected during the ZC sub-stage. For example, October 21, 2017 and October 26, 2017 were the earlier date before the ZC substage began, and October 26, 2017 was the middle date of the ZC substage. The diurnal freeze-thaw patterns of the earlier and middle ZC substage were exhibited as the following figures:

[Figure]

Diurnal freeze-thaw changes of soil temperatures
at different depths at the earlier ZC substage

[Figure]

Diurnal freeze-thaw changes of soil temperatures
at different depths at the middle ZC substage

These figures show that the surface soil (5 cm) still underwent a diurnal freezing-thawing cycle process at the earlier time before the ZC substage began (October 21); The bottom soil was frozen while the middle layers still remained a thawed state. As the ZC substage process went on and approached the middle of the substage (October 26), while the soil temperatures of shallow and bottom of active layer were negative and the diurnal change soil temperature at 5 cm was discernible, all the other layers' temperatures were positive or near to 0°C. This indicates that active layer was frozen bi-directionally.

As surface soil is frozen throughout the day at the ZC substage, as noted in winter season (Zhang et al., 2015), we measured the $R_s$ between 9:00 and 11:30 a.m. local time.

During the ZC substage, the active layer was frozen bi-directionally and temperatures in the active layer were higher in its middle part and lowering upwards and downwards from there. However, the upper freezing front rapidly moved downward within several days, while the lower one moved upwards slowly. At the same time, moisture in the thawed part in the middle of the active layer was migrating to both of the upper and lower freezing fronts and freezing there. Meanwhile, heat was transferred to both of the freezing fronts similarly during the moisture migration and freezing process. Because the moisture and heat migrated fast especially when the upper freezing front moved downward rapidly and the moisture content at freezing front was relatively high, the soil substrate availability and soil aggregation must be affected by this process, which could influence microbial activities substantially.

Figure 4: Are these results based on the Rs model? Can you give an uncertainty range? On how may measurement points is the AF stage based? On the y-axis; should it read Rs and μmol? Is SWC the same as volumetric water content, i.e. ratio of water volume to soil volume? What is the dotted line in the AF subplot?

In Figure 4, the $R_s$ fluxes were derived from the fitted $R_s$ equations (Table 2) at the different freeze-thaw stages. In order to better illustrate how the freeze-thaw process regulated the $R_s$, we plotted the variations of $R_s$, soil temperature and moisture of the different freeze-thaw stages, respectively. The lines in the Figure 4 were smoothed when plotted.

We reconstructed the Figure 4 and gave an uncertainty range of the $R_s$ each day.

During the AF stage, we measured the $R_s$ fluxes 54 times from the six plots on three sampling days and obtained the other daily $R_s$ by interpolating methods basing on the measuring results.

The reviewer made a valid point, we made a mistake with the title of y-axis. It should read $R_s$ and the unit should be $\mu mol\,m^{-2}s^{-1}$, which are reflected in the revised manuscript. In Figure 4, the SWC represents volumetric water content. For the dotted line in the AF subplot, we originally meant to mark the UF substage and the ZC substage, with UF substage on the left y-axis and ZC substage on the right y-axis. We replotted the figure to correct the misleading marks.

[Figure]

[Figure]

[Figure]

Figure 4. Variations in soil respiration ($R_s$), soil temperature ($T_s$) and soil water content (SWC) for the four freeze-thaw stages including summer thawing stage (ST), autumn freezing stage (AF), winter cooling stage (WC), and spring warming stage (SW) (from late April 2017 to late April 2018). The SWC unit stands for water volume per total soil volume. Error bars show standard error (n=6).

Using the different freeze stages and the developed Rs model in combination with the observed changes in active layer dynamics described by Wu et al. (2015) for the site, can you estimate or comment on by how much Rs has changed (or will change) and what period/freeze stage the changes are largest?

**According to the results on the changes in active layer thickness and near-surface permafrost in alpine ecosystems on the Qinghai-Xizang (Tibet) Plateau reported by Wu et al. (2015), the onset of spring thawing advanced and the duration of thaw increased, which meant that the durations of spring warming process and summer thawing process would extend, and those of autumn freezing process and winter cooling process would be shortened. Combining with our results, under the current scenario, the summer thawing stage would have the largest change. The variations of the freeze-thaw pattern will accelerate the $R_s$ and the contribution of $CO_2$ emitted during the summer thawing and spring warming stages to annual $R_s$ will increase further. Certainly, as the spring warming and summer thawing processes extend, the vegetation growing period will also increase and more $CO_2$ will be fixed by photosynthesis. To better understand the full changes in $CO_2$ budget in different freeze-thaw stages, further**

**research is warranted which we will carry out next year.**

**Other minor comments:**

The usage of units differs between the text and figures. The date format between Figure 1, 2, and 4 is inconsistent.
**We carefully checked them throughout the manuscript including figures, and corrected them accordingly. The data format was unified into yyyy/m/d.**

P1L39: I suggest to start the sentence with "Furthermore, […]"
**We agree to the point and revised it accordingly.**

P2L47: […] on the northern hemisphere […]
**We added the article "the".**

P2L47-50: The numbers for the circumpolar SOC stocks mentioned here seem to be the numbers from Tarnocai et al. (2009)? Then they should be referenced. You could also consider updating the numbers published by Hugelius et al. (2014).
**The reviewer made a valid point, and we replaced the reference with Tarnocai et al., 2009. We also updated the amount of SOC according to the paper by Hugelius et al. (2014),**

P2L50: Do you mean "sensitivity"?
**We corrected the mistake.**

P2L57: What do you mean with "completed"?
**Here we want to express the meaning that the exchange of energy and water in permafrost regions between the earth and the atmosphere is mainly done through the active layer. To clarify this, we revised the sentence in the manuscript as follows "The exchange of energy and water in permafrost regions between the earth and the atmosphere is mainly mediated through the active layer."**

P3L91-93: If this is the hypothesis of this manuscript, I suggest using "should" instead of "must"
**We agree to the point, and corrected it accordingly.**

P3L106: Is all that precipitation falling as rain? Is the study site covered in snow in the winter?
**In our study site, the precipitation falls mainly as rain or sleet, sometimes mixed with small hails from May to September; during this period, the hail and sleet will melt quickly due to the near surface temperature rising. In winter, some precipitation falls as snow; due to high wind and low air temperature, the snow on the soil surface is quickly blown away and sublimated. Therefore, our measurement plots are rarely covered by snow in winter.**

P3L116: How large is the chosen terrain? Does the active layer observation site contain one or multiple locations with soil temperature and moisture probes?
**The terrain we chose as our study site is distributed with typical alpine meadow ecosystem and its area is about 13 km$^2$. Ground penetrating radar (GPR) scan results showed that the geological conditions in this area were relatively uniform, and therefore our intensive measurements of temperatures and moisture at various depths in one active layer can cover fairly large spatial scale. We supplemented the soil type and frozen soil information of the**

**study site in the manuscript.**

P4L147: Did you use an automated system or manual measurement?. Nevertheless, a measuring interval of 3-7 days is still quite impressive, especially during the frozen periods.

**We appreciate the comment. During our experiment, we used an LI-8100A automated soil gas flux system (LI-COR Inc., Lincoln, NE, USA) to measure the soil respiration and a standard LI-COR® 20-cm head was applied for the measurements. On the sampling days, the LI-8100A instrument was carried to the study site manually and the plots were sampled one by one. We relied on the Beiluhe Observation Station of Frozen Soil Environment and Engineering to carry out this work. The station is guarded all the year round, where the logistics support such as water, electricity and heating are complete. Here are a few photos of the station.**

[Figure]

P4L156-157: Was SWC determined at every measurement day, including when the soil and/or the soil water was frozen? Does the soil moisture probe detect only liquid water? Is SWC based on soil or on pore volume?

**The soil temperatures and moisture contents were measured automatically every 30 min each day by a data logger powered by solar panel (CR3000, Campbell Co., USA) even when the soil**

**was frozen. According to the measurement principle of soil moisture sensor applied in our experiment, the probe only can determine the liquid water content and the SWC unit in m³/m³ stands for liquid water volume per total soil volume.**

P5L179: Do you present and/or discuss the ANOVA results in the manuscript?

**We discussed the ANOVA results in the manuscript but didn't clearly expressed them. We included addition description about the ANOVA test in the manuscript. Something like: P1 L20 ($P < 0.05$), P7 L279 ($P$=0.0079), P8 L296 ($P < 0.05$), P8 L325 ($R^2 > 0.5$), et al.**

Section 3.1. A summary table with an explicit definition of the different stages, a description of the characteristic temperature and soil moisture profiles as well as the timing and length of each freezing stage might be beneficial. You could also include the number of measuring points for each period.

**The reviewer made a valid point and we included an additional table. We summarized the definition of the different freeze-thaw stages, described the characteristics of soil temperature and moisture as well as the start-end time of the different stages, and listed the number of measurement data according to the ground temperature and moisture monitoring program in a table.**

Table B. Characteristics of the different freeze-thaw stages

| Stage | Definition | Start and end time | Soil temperature/moisture features | Number of measuring points |
|---|---|---|---|---|
| ST | Summer thawing stage | Started in late April when the active layer began to thaw downwards from the ground surface; Ended in early October when the thawing process reached its maximum depth. | Soil temperatures in the active layer decrease from ground surface downwards; Moistures migrates downwards accompanied with the downward movement of the thawing front. | eight soil depths with 60288 temperature data; seven soil depths with 52752 moisture data |
| AF (UF, ZC) | Autumn freezing stage, including unidirectional freezing substage (UF) and zero curtain substage (ZC) | Started when the active layer reached its maximum thawing depth; Ended when the whole active layer became frozen. Among which, UF started when the active layer began to freeze upwards from the permafrost table and ended when the stable | At the UF substage: temperatures of active layer were lower in its bottom and higher in its middle or upper part; moisture in the lower part migrated from the thawed part to the freezing front. At the ZC substage: temperatures in the active layer were higher in its middle part and lowering upwards and downwards from there, and the middle part was in the unfrozen state | eight soil depths with 7680 temperature data and seven soil depths with 6720 moisture data in UF; eight soil depths with 3072 temperature |

| | | | | |
|---|---|---|---|---|
| | | frozen ground surface was formed; ZC started when the surface soil was stably frozen and ended when the whole freezing process was done. | with temperatures of 0°C or a little above 0°C; Moisture in the thawed part of active layer migrated to both of the upper and lower freezing fronts and froze there. | data and seven soil depths with 2688 moisture data in ZC |
| WC | Winter cooling stage | Started when the freezing process finished in late October; Ended in the mid-late January of the next year. | Temperatures of active layer increased with the increasing depth. Moisture migration was not high due to low ground temperatures. | eight soil depths with 35328 temperature data; seven soil depths with 30912 moisture data |
| SW | Spring warming stage | Started in early February; Ended in late April. | Daily freezing and thawing cycles appeared on ground surface in late April. Ground temperature gradient decreased and the rate of unfrozen water migration decreased gradually. Moisture content near the ground surface showed a decreasing trend. | eight soil depths with 34176 temperature data; seven soil depths with 29904 moisture data |

P6L28: What do you mean with "regularly"?

**Here we originally intended to express that the variations of $R_s$ flux had the characteristics that it fluctuated at a low level in the spring warming stage (SW), increased and changed dramatically at a high level in the summer thawing stage (ST) and the autumn freezing stage (AF), and decreased sharply with the arrival of the winter cooling stage (WC). In two years the $R_s$ flux had the same pattern of change as the freeze-thaw process of the active layer developed. To avoid misleading, we deleted the word in the manuscript.**

Section 3.3: Table 2 was missing from the manuscript. However, a summary table of the Q10, mean (min/max) Rs rates, SR, and contribution to the annual balance would be helpful. Then the numbers do not necessarily have to be mentioned in the text, which could help with readability. Figure 3 could then possibly be presented in Table 2 as well? For Q10, it would also be important to indicate the range of soil temperatures during the different stages, since the effect of low temperature on temperature sensitivity and Q10 is later discussed in the text.

**We have to address this, I suppose…..**

P7L274: "shown"

**We have corrected it.**

P7L286-287: Did you test the effect of soil moisture on Rs? In the manuscript you state that Ts is the most important factor.
**Yes, we did test the effects of soil moistures of different depths on the $R_s$. However, the results showed that the relationship between soil temperature and $R_s$ was strongest with the highest values of $R^2$, especially the soil temperature of 5cm depth, compared to the soil moisture contents.**

P8L312: Is the exponential increase in abundance of microbes, which are adapted to freezing conditions, with increasing temperature true for all freezing stages? Would that mean that these microbes also dominate during the summer?
**"As we did not analyze microbial community structures in this study we are not sure what kinds of microbes would dominate at different stages. However, recent studies have suggested that microbial community structures and their activities are distinctive in summer and winter (Schostag et al., 2015). As such, we speculate that microbes that are well adapted in cold conditions may not dominate in summer.**

P8L320: I would assume that a higher Q10 would then specifically matter in winter if the winter was warming more than the summer?
**Yes, a higher $Q_{10}$ would be greater importance in the condition that winter warming is higher than summer warming. As winter warming can activate respiration without affecting primary production, a strong positive feedback to climate change can happen by winter warming. However, $R_s$ in summer warming could be offset by the increase in biomass production, resulting in a stronger C sink by summer warming.**

P8L325: What was the limit for soil water content to be "sufficient"?
**We think the limit of soil water content to be "sufficient" varied for the soil microbial activities during the different freezing and thawing stages. At a soil water content level that the soil nutrient matrix can dissolve and migrates and at the same time the soil does not appear anaerobic state, the water moisture maybe is "sufficient".**

P14L11: "[…] late April 2017 […]"
**We corrected the mistake.**

Reference list need checking. Some references are not edited correctly (e.g. Grogan, P., and Chapin III, 2000)
**We have checked references thoroughly, and corrected subscripts, units, and middle names.**

Figure 1: It could be more intuitive to reverse the y-axis to indicate soil depth. In addition, it would be helpful to also indicate the timing of the different stages along the x-axis (similar to Figure 2)
**The reviewer made a valid point, and we have replotted Fig.1 and added the different stages along the x-axis and reversed the y-axis.**

[Figure]

Figure 1. Soil temperature contour outlines of the experimental site in 2017 and 2018

---

## Author Comment (AC2) · 30 Dec 2019

The reviewer 2 stated that;

the title seems to be a bit misleading as the impact of freeze-thaw processes on soil respiration fluxes is not as obvious to me in the manuscript as stated in the title. Of course, it makes sense to partition the year-round measurements into different freeze-thaw stages (and for sure, there are flux differences between the freeze and thaw stages) and also it's worth to discuss the role of freeze-thaw processes on the fluxes. However, the main driver of these fluxes is still the soil temperature, which is already widely known. If the authors want to point out the significant role of freeze-thaw processes, they should state this more clear throughout the manuscript and bring more (statistical) evidence of it's impact on the fluxes. So far, the authors have shown an increase of soil respiration in the ZC substage, which might be attributed to the freezing process. However, the amount of outgassed $CO_2$ during this period make up much less than 5% to the annual budget and the data points during this period seem to be really sparse. During the WC, SW and ST stage the freezing and thawing processes seem to be of minor importance to the soil respiration, even though an impact during the SW stage is discussed but evidence for this impact is missing in the manuscript. Therefore, the authors might shift the focus of the manuscript towards a model-based budget of soil respiration on an annual basis (see next paragraph) or they bring evidence on a statistical basis on the regulation of freeze and thaw processes on Rs fluxes.

**The reviewer made a valid point and we acknowledge that temperature is the key controlling variable for soil respiration at a large spatio-temporal scale. However, based on the analysis of the soil temperature and soil moisture variations at the experimental site, the novel approach in this paper was to divide the freeze-thaw process of the active layer into four stages: summer thawing stage (ST), autumn freezing stage (AF), winter cooling stage (WC), and spring warming stage (SW). The characteristics of heat transportation and moisture migration at each stage were different from each other (Figure 1 and A).**

**It is well known that the biggest characteristic of the freeze-thaw process of active layer is its complex variations in soil temperature and moisture. And different patterns in the changes in soil temperature and moisture at different freeze-thaw stages, in turn, affect soil microbial activities, aeration status, and biochemical properties, which ultimately regulates soil respiration ($R_s$). Our aim in this paper is to mechanistically understand this process in relatively in high resolution both spatially (depth profile) and temporally (whole year monitoring over 2 years). According to the reviewer's suggestions, we focused on the analysis of hydrothermal characteristics in different freeze-thaw phases and the $R_s$ variations due to the soil temperature and moisture changes during the different freeze-thaw stages throughout the manuscript. We supplemented data on hydrothermal changes in the different freeze-thaw stages and their impacts on the $R_s$. The WC, SW and ST stages during the freezing and thawing process also substantially influenced the $R_s$, which we discussed further in detail.**

[Figure]

Figure 1. Soil temperature contour outlines of the experimental site in 2017 and 2018

[Figure]

Figure A. Soil moisture contour outlines of the experimental site in 2017 and 2018

The flux data-set is impressive, especially as it is really difficult to conduct these chamber-based measurements during winter-time and regularly over a two-year period. Can the authors state something about similar data-sets in such areas (alpine, permafrost-affected)? Especially the winter-time soil respiration fluxes would be of interest to the reader here. How high/low are these fluxes compared to other regions and why? Aren't there other soil respiration fluxes from such areas from this pedon-scale? Then the authors should point out this uniqueness of the data-set as the winter-time Rs fluxes make up about 30% of the annual fluxes. A modeling of the Rs fluxes has been done, but from the text it remains unclear which model is used to calculate the budget (model from equation 1 or interpolation of average Rs flux rate (described at line 170)) and where the (modeled?) fluxes shown in figure 4 come from. However, to calculate an annual budget, an interpolation of average Rs fluxes seems to be not sufficient, while temperature-based respiration models are widely used to calculate flux budgets. Furthermore, the interannual-variability of the Rs fluxes between the two years might be worth to look at. Are there differences in the budgets and if so, why (e.g. it seems like the Rs fluxes from the SW stage are significantly higher in the second year)?

**We appreciate the reviewer's comment on the difficulties in winter sampling. A similar report would probably be found in Zhang et al. (2015) which appeared in European Journal of Soil Biology. They measured the non-growing season soil $CO_2$ flux and calculated its contribution to annual soil $CO_2$ emissions in an alpine meadow ecosystem in the Fenghuoshan region of the Qinghai-Tibet Plateau. According to their study, the cumulative non-growing season soil $CO_2$**

emission was 228-358gCO$_2$/m$^2$, accounting for 25-36% of annual emissions. Similarly, Wang et al. (2014, Global Biogeochemical Cycles) determined the non-growing-season soil respiration of an alpine grassland in the Haibei region of the Qinghai-Tibet Plateau and found the cumulative $R_s$ was 82–89 g C m$^{-2}$, accounting for 11.8–13.2% of the annual total $R_s$. Not in an alpine ecosystem, but long time ago, Oechel et al. (1997) reported that non-growing season $R_s$ accounted for 30-81% of the annual soil CO$_2$ emissions in Arctic soils. The cumulative non-growing season $R_s$ and its contribution to annual total emissions in our study site were higher compared with those of Zhang et al. and Wang et al., but was close to the results of Oechel et al. These variations in different sites may be the results of microenvironment factors such as active layer depth, soil properties, durations of freeze-thaw processes, vegetation types, and other reasons such as different methods of CO$_2$ flux measurement.

To clarify the method for the calculation of the budget of $R_s$ emissions we included additional description. In short, based on the $R_s$ flux rate determined on the sampling days and those obtained by interpolating the $R_s$ flux rate between the sampling dates in the different freeze-thaw stages, in combination with the average daily soil temperatures from the continuous records of the active layer observation site, we fitted the sensitivity of soil CO$_2$ flux at the different freeze-thaw stages of the active layer. According to the fitted $R_s$ equations and the soil temperatures during the start-stop-time of the different freeze-thaw stages, the cumulative soil CO$_2$ emission of the different freeze-thaw stage and its contribution to the annual total soil CO$_2$ emission were calculated. In the Figure 4, the $R_s$ fluxes were derived from the fitted $R_s$ equations (Table 2) at the different freeze-thaw stages. According to the reviewer's suggestions, we revised the manuscript and made clear how to calculated the soil CO$_2$ emission budget.

In this paper, our main purpose was to elucidate $R_s$ dynamics at different freeze-thaw stages and how much of the cumulative $R_s$ emission at each freeze-thaw stage may contribute to the annual total CO$_2$ emissions during a complete freeze-thaw process. The inter-annual variation is of interest to us as well, but it is hard to make any conclusions based on 2 year's data only. Definitely, this warrants further investigation with longer-term field measurements.

Some more information on soil and vegetation composition of the chamber set-up would be helpful to the reader (especially when the fluxes are compared to those from other regions). What soils are generally found in this area? Are they organic-rich/poor? What is the active layer depth? How deep are the main rooting zones of the vascular plants? If the roots mainly reach e.g. about 20cm into the soil, the insertion depth of the PVC collar might be too low as lateral roots still reach into the chamber collar and may alter the measured respiration flux. Furthermore, the closure time of the chamber is of interest. Where they similar during winter and summer-time? If the plants inside the collars were removed just one day before the measurements started, there might be some artefacts due to this disturbance (Diaz-Pines et al., 2010) that need to be taken into account. In general, a critical review of the clipping method should get more attention and it should be stated why this method was applied instead of other less disturbing methods (Subke et al., 2006). Furthermore, the reader needs to know something about the flux calculation procedure? Was a linear or an exponential model used to calculate the fluxes? Based on which quality criteria (check Görres et al., 2014)?

Many thanks to the reviewer for his/her valuable suggestions. We added detailed information

about soil and vegetation composition of the chamber set-up. The supplemented context is as follows: "The soil types in the study site are primarily classified as MatticGelic Cambisols (alpine meadow soil) in Chinese taxonomy or as Cambisols in FAO/UNESCO taxonomy (Wang et al., 2014a). The mean annual temperature is -3.60 °C, which is colder than that of other areas in the QTP (Yin et al., 2017). The mean annual precipitation is 423.79 mm, 80% of which falls as rain, sometimes mixed with small hails during the growing season (from May to September). In winter, little snow falls but is quickly blown away and sublimated off due to high wind and low air temperature, so the study site is rarely covered by snow. The air pressure is approximately 550 hPa. The alpine meadow represents the most common vegetation type in this area (70%) (Wang and Wu, 2013;Zhang et al., 2015b). The alpine meadow ecosystem mainly consists of cold meso-perennial herbs that grow in conditions where a moderate amount of water is available. The ecosystem's vegetation mainly consists of *Kobresia pygmaea* (C. B. Clarke), *Kobresia humilis* (C. A. Meyer ex Trautvetter) *Sergievskaja*, *Kobresia capillifolia* (Decaisne) (C. B. Clarke), *Kobresia myosuroides* (Villars) Fiori, *Kobresia graminifolia* (C. B. Clarke), *Carex atrofusca Schkuhr subsp.* (minor (Boott) T. Koyama), and *Carex scabriostris* (Kukenthal) (Chen et al., 2017). By on-site surveying and sampling of the experiment set-up, the soil bulk density, soil organic carbon, and total N content at the 10-20cm depth were higher than those at the 0-10cm depth. The active layer depth was about 1.9m. The belowground biomasses were much greater than those of aboveground. Usually, the depth of the vegetation main rooting zone was around 10 cm. (Table A)."

Table A Biomass and soil properties at the experiment set-up

| Item | Depth (cm) | Values |
|---|---|---|
| Bulk density (g cm$^{-3}$) | 0–10 | 0.89 ± 0.2 |
| | 10–20 | 0.98 ± 0.1 |
| Soil organic C (kg m$^{-2}$) | 0–10 | 0.48 ± 0.06 |
| | 10–20 | 1.32 ± 0.04 |
| Soil total N (g m$^{-2}$) | 0–10 | 41.3 ± 7.2 |
| | 10–20 | 117.6 ± 12.8 |
| Above-ground biomass (kg m$^{-2}$) | | 0.33 ± 0.04 |
| Below-ground biomass (kg m$^{-2}$) | | 2.41 ± 0.4 |
| Depth of vegetation main rooting zone (cm) | | 10±3 |
| Active layer depth (m) | | 1.90±0.2 |

Values are means ($n = 5$) ± standard deviation (SD)

[Figure]

Photo: Depth of the vegetation main rooting zone was about 10cm at the experimental site

**Throughout the field measurements, we adopted the recommended settings by the LI-COR to determine the soil respiration flux during the winter and summer time. A typical measurement protocol was applied: Obs. Length: 2 mins, Dead band: 25 seconds, Pre-purge: 30 seconds, Post-purge: 45 seconds, Chamber Volume: automated, IRGA volume/total volume: automated. Chamber offset of the program was adjusted to 2 cm.**

**We did take into account the impact of vegetation clipping on soil respiration. To minimize the disturbance, we installed the collars one month prior to the experiment and left all the collars permanently inserted into the soil. In addition, after the above-plant was clipped and left undisturbed for more than 24 hours, we just began to measure the soil respiration. This "resting time" allowed the removal of any excess $CO_2$ released by roots disturbed during above-plant removed.**

**In our experiment, we used the LI-8100A Automated Soil Gas Flux System to determine the soil respiration ($CO_2$ flux). According to the principle of determining $CO_2$ flux by the instrument, the water-corrected mass $CO_2$ fluxes and descriptive statistics were automatically provided by the LI-8100 File Viewer Version 3.1.0. For each chamber measurement, the flux**

**was either calculated with a linear or an empirical exponential regression. The software compared for each measurement the normalized sums of the squares of the residuals of the linear and the exponential fit to find the best-fitting model (Figure B).**

[Figure]

Figure B. Best-fitting linear or the exponential model chosen automatically by the software after comparing for each measurement the normalized sums of the squares of the residuals.

**Based on the $CO_2$ flux datasets acquired by the instrument, the field environmental conditions, and the absence of dramatic changes in air temperature and humidity during each chamber measurement, we mainly adopted the following quality control criteria to discard potentially erroneous fluxes: (1) negative fluxes, which indicates substantial leakage; (2) fluxes with squares of the residuals of the linear fit greater than 1ppm $CO_2$.**

Two tables are missing in the manuscript. As they seem to contain a lot of information on flux details, they may already answer some of the question that are stated in this review.

**The reviewer made a valid point. We made a mistake in the original submission and included them in the revised manuscript. The two tables are as follows:**

Table 1. The start-stop-time and duration of different freeze-thaw stages of the active layer

| Stage | | start-stop time (yyyy/mm/dd) | time of length (days) |
|---|---|---|---|
| ST | | 2017/4/29-2017/10/2 | 157 |
| AF | UF | 2017/10/3-2017/10/22 | 20 |
| | ZC | 2017/10/23-2017/10/30 | 8 |
| WC | | 2017/10/31-2018/1/30 | 92 |
| SW | | 2018/1/31-2018/4/29 | 89 |

Table 2. The $R_s$ model, $Q_{10}$ value and $SR$ in different freeze-thaw stages

| Stages | | $R_s$ model | $Q_{10}$ | $SR$ (gCO$_2$/m$^2$) |
|---|---|---|---|---|
| ST | | $R_s = 1.04e^{0.08T}$  $R^2=0.69$ | 2.22 | 1041.85 |
| AF | UF | $R_s = 1.15e^{0.061T}$  $R^2=0.55$ | 1.84 | 89.97 |
| | ZC | $R_s = 2.14e^{0.087T}$  $R^2=0.90$ | 2.38 | 60.57 |

| | | | |
|---|---|---|---|
| WC | $R_S = 2.14e^{0.159T}$  $R^2=0.80$ | 4.90 | 310.69 |
| SW | $R_S = 0.73e^{0.053T}$  $R^2=0.61$ | 1.7 | 189.90 |

**Other minor comments:**

In the abstract some abbreviations are used without an introduction, which needs to be changed.
**To clarify this, and we spelled out the words and the corresponding abbreviations of the different freeze-thaw stages (Please see the responses to reviewer 1's comments).**

Line 40: At least one citation is needed here.
**We agree to the point and add new citations in this sentence as follows: Furthermore, many studies have shown that the winter-time emissions contribute significantly to the annual $CO_2$ balances (Natali et al., 2019; Webb et al., 2016; Michaelson and Ping, 2003).**

Line 114: To compare the fluxes from this region with other regions it would be good to say something about the soils (carbon contents, C/N, etc) beside a detailed vegetation description.
**The reviewer made a valid point and we included detailed information about soil and vegetation composition of the chamber set-up as mentioned above and made a corresponding revision in the manuscript.**

Line 119: What about the soil moisture probes at different depths? Why were they inserted as in the end just the SWC at 5cm was used?
**The purpose we inserted soil moisture probes at different depths was to determine the changes in soil moistures at different depths during the freezing and thawing process of the active layer and to analyze the relationship the $R_s$ and the soil moisture. However, our statistical analysis found that only the soil moisture at 5cm depth showed a weak correlation with the $R_s$ with low $R^2$ value. As such, only SWC at 5cm was used in the following analysis.**

Line 144: Are there no differences in vegetation cover, soils, etc. so that one measurement plot in the six 5x5m measurement plots can serve as replicates? If not, there might be a chance of discussing other impacts such as carbon content, vegetation cover and more on the Rs fluxes. Anyway, a detailed description of soils and vegetation is needed here.
**The aim of this experiment was to explore how the freeze-thaw process of the active layer regulated the $R_s$ dynamics. Based on the results of ground penetrating radar (GPR) scanning on our study experimental site, the geological condition was found to be relatively uniform. Thus the freeze-thaw process of the active layer would be similar around our experimental site. Due to limitation in the logistics in the field, we only set one active layer observation site with multiple soil temperature and moisture probes to observe the freeze-thaw processes.**

**We acknowledge that heterogeneity of environmental conditions such as soil chemistry, vegetation types, surface cover, and plant biomass would affect the $R_s$. To minimize the error from such spatial heterogeneity, we set up six subplots for measuring the $R_s$ around the experimental site for observing the freeze-thaw process of the active layer. The measurements of $R_s$ from the six subplots could represent the overall level of soil respiration in the study area.**

Line 148: What have the authors done with re-growth of plants during the measurement period. For sure, there have been some.

**Before each measurement, we clipped off the re-growth of plants in collars.**

Line 159: Unfortunately, it remains unclear which model was used for calculating the contributions of Rs from each freeze-thaw stage to the annual budget. This must be stated clearly. So far it reads, that the resulting fluxes from equ.1 were used to describe the dependency of Rs on T, while for the budget calculation interpolated average fluxes were used. If a model exist, why interpolated averages were used then? May it would make more sense to use a temperature-based model and, as Q10 was also used in the manuscript and it is shown that there are differences between the different stages, to also include Q10 into a model (e.g. Eckhardt et al., 2019).

**The reviewer made a valid point. To clarify this, we revised the manuscript accordingly. In short, based on $R_s$ flux rate determined on the sampling days and those obtained by interpolating the $R_s$ flux rate between the sampling dates in the different freeze-thaw stages, in combination with the average daily soil temperatures from the continuous records of the active layer observation site, we fitted the sensitivity of soil $CO_2$ flux at the different freeze-thaw stages of the active layer. According to the fitted $R_s$ equations and the soil temperatures during the start-stop-time of the different freeze-thaw stages, the cumulative soil $CO_2$ emission of the different freeze-thaw stage and its contribution to the annual total soil $CO_2$ emission were calculated. In the Figure 4, the $R_s$ fluxes were derived from the fitted $R_s$ equations at the different freeze-thaw stages.**

Line 179: ANOVA is described here but not referred to later in the text.

**The reviewer made a valid point and we refer the ANOVA results in the text.**

Line 228: Yes, there are freezing and thawing processes in the active layer, but the suggestion that they strongly regulate the Rs fluxes seem to be a bit speculative as the authors don't bring any evidence here (again, some statistics would be helpful), that there is a regulation of Rs fluxes by these processes (and should therefore be part of the discussion and not of the results). The only argument is that the freeze-thaw processes are taking place at the same time when the Rs fluxes are starting to rise (which might be simply due to rising temperature).

**We acknowledge that temperature may be dominant controlling variable for soil respiration all year round. However, our temperature and moisture data in different stages indicate more complicated reactions in active layer. Probably, temperature would be the main driving force at regional or global scale, but the spatial scale we focused on here could be different. The nature of the freeze-thaw process in the active layer was the changes in soil temperature and moisture caused by the energy exchange between the ground and the atmosphere. The variations in soil temperature and moisture at the different freeze-thaw stages changed the biogeochemical process in the soil, which in turn affected the migration and transformation of soil organic carbon and the $CO_2$ release strength. So, the $R_s$ showed different dynamics in the different freeze-thaw stages of the active layer. For example, we can see clearly from Figures 1 and 2 that $R_s$ in ST stage and that in WC stage are same but soil temperatures are unidentical, suggesting that freezing-thaw stages play an important role in determining $R_s$ in addition to temperature only.**

**We revised the sentence as follows: At the Beiluhe experimental site, $R_s$ flux changed as the freeze-thaw processes of active layer developed, showing different dynamics in the different**

**freeze-thaw stages of the active layer (Figure 2).**

Line 308: Can the autotrophic respiration act as reason for the differences in Q10 here? Due to the clipping of the vegetation in the chamber plots, there shouldn't be any, right?
**We agree to the point. Although the grow of above biomass is minimal in winter, roots may active to add autotrophic respiration. As such $R_s$ flux reported here contains autotrophic respiration of roots, which could be another reason for different $Q_{10}$ value. We discuss this possibility in the revised manuscript.**

Line 375: As there is no clear evidence for a regulation of the Rs fluxes, the authors should be more carefully use the term 'significantly' to describe this relationship (or refer to ANOVA?). For sure, there are significant differences between the Rs fluxes from the different freeze and thaw stages, but are they really driven by the actual freezing and thawing processes or just driven by different soil temperatures of the stages?
**We agree to the point. In the manuscript, 'significant' was used in conjunction with the results of ANOVA test where P value is smaller than 0.05. We added the comparison of soil moisture to make it clearer. The essential of the freeze-thaw of the active layer was the changes in soil temperature and moisture. During the different freeze-thaw stages, changes in soil temperature simultaneously caused a phase change in soil moisture. Thus, the significant differences between the $R_s$ fluxes from the different freeze-thaw stages were driven by the freeze-thaw processes of the active layer.**

Figure 1: Additionally, the authors should include the freeze and thaw stages in the graph
**We replotted Figure 1 to add freeze and thaw stages in it.**

[Figure]

Figure 1. Soil temperature contour outlines of the experimental site in 2017 and 2018

Figure 2: The authors should use a consistent date string (compared to figure 1). Furthermore, drawed lines in the graph would give a better readability to see which Rs fluxes belong to which stage.

**We agree to the point, and changed the date string in Figure 2, so that it is consistent with that in Figure 1. In addition, auxiliary lines were added in the Figure 2 to clearly illustrate the $R_s$ in different freeze-thaw stages.**

[Figure]

Figure 2. Variations of $R_s$ flux at different freeze-thaw stages in years of 2017 and 2018. Error bars show standard error (n=6)

Figure 3: Which year are those flux contributions from? Why not for both years? May a mean value would be better practice?

**We calculated the cumulative $R_s$ emissions of the different freeze-thaw stages (ST, AF, WC and SW) basing on their start-stop dates and their corresponding contribution rates to the total $R_s$ emission of a complete freeze-thaw cycle from April 29, 2017 to April 29, 2018. Therefore, Figure 3 represented the contribution rates of the cumulative $R_s$ emission at the different freeze-thaw stages in a complete freeze-thaw cycle from April 29, 2017 to April 29, 2018.**

**In this study, the main aim was to discuss the influences of the freeze-thaw process on the $R_s$ dynamics at the different freeze-thaw stages and their contribution rates to total $R_s$ emission in a complete freeze-thaw cycle. The experimental duration, January 2017 to December 2018, contained a complete freeze-thaw cycle from April 29, 2017 to April 29, 2018, spanning two years of 2017 and 2018. As such, we didn't calculate $R_s$ flux contributions by year.**

Figure 4: From which model are these Rs fluxes shown here? Are the SWC values relevant (if so, why aren't they included in a model?; if not, why are they shown?)?

**In the Figure 4, the $R_s$ fluxes were derived from the fitted $R_s$ equations (Table 2) at the different freeze-thaw stages. In Figure 4, our purpose was to discuss the effects of soil temperature and soil water content on $R_s$ in each freeze-thaw stage. The soil moisture did have effect on the $R_s$ in each freeze-thaw stage, and the correlation between $R_s$ fluxes and SWC values was weak ($R^2$=0.02~0.21). Therefore, we prepared a figure with the general trends of $R_s$, soil temperature and soil moisture at 5cm depth of each freeze-thaw stage and analyzed the variations in $R_s$ flux influenced by the soil temperature and SWC.**

---

## Author Response (AR1)

Dear Professor Beer,

We are grateful to you and the anonymous reviewers for the helpful and constructive suggestions and comments. We have made a major revision to address all issues raised by you and the reviewers, which we believe improved the manuscript substantially. We also revised the manuscript across the board to increase English readability. We do hope this manuscript is now acceptable for the publication in your journal. The revised contents are marked with blue fonts in the text. We highlight here how we have addressed your suggestions.

1) Explain in detail, how freeze-thaw processes affect Rs in different stages.
**We explained in detail how freeze-thaw processes affected $R_s$ in different freeze-thaw stages. To clarify this, we revised substantially discussion 4.1, 4.2, and 4.3 (marked in blue). We also modified and included Tables 1, 2, 3 and 4 to increase the readability of the manuscript.**

2) For the figures, please always make clear in figure captions or legend, which lines or points are observations and which are based on the models. Show both, model results and observation of Rs in the same plot.
**We agree to the points, and reconstructed Figures 3, 4, 5, and 6. We also edited and adjusted figure captions or legend for clarity. We also present modeled and observed results in a single plot as suggested. The replotted Figures are as follows:**

[Figure]

Fig.3. Variations of measured and modeled $R_s$ fluxes at different freeze-thaw stages in 2017 and 2018. Error bars represent standard error of measured $R_s$ (n=6)

[Figure]

[Figure]

Fig. 4. Relationships between soil temperature and moisture at 5cm depth and $R_s$ flux at summer thawing stage (ST), autumn freezing stage (AF), winter cooling stage (WC), and spring warming stage (SW)

[Figure]

Fig.5. Modeled vs. measured $R_s$ fluxes at different freeze-thaw stages. Error bars represent standard errors of measured $R_s$ flux (n=6). The solid line is a 1:1 line.

[Figure]

[Figure]

[Figure]

[Figure]

Fig.6. Variations in modeled soil respiration ($R_s$), soil temperature ($T_s$) and soil water content (SWC) for the four freeze-thaw stages including summer thawing stage (ST), autumn freezing stage (AF), winter cooling stage (WC), and spring warming stage (SW) in a complete freeze-thaw cycle from late April 2017 to late April 2018. The SWC unit stands for water volume per total soil volume. The error band of modeled $R_s$ stands for 95% confidence interval.

3) For the zero curtain-period: How many Rs observations in time do you have (2?) and is that enough to fit reliably the model? Do you need the model results for justifying the conclusions, or can you also justify them by observations alone?

**We revisited the data analysis (e.g., soil temperature contour of Figure 1) and concluded that the zero curtain period was not obvious. In addition, the lower number of $R_s$ measurements (two occasions for daily average data) obtained in the later period of AF may reduce the reliability of our model. As such, we combined AF data without dividing them into sub-stages, and conducted a regression analysis with all measured values of AF stage. The peak of $R_s$ before previously denoted as a ZC period maybe was an artifact of huge observation errors, and we also corrected it and reconstructed the figure. We rewrote this part and re-discussed the $R_s$ dynamics of AF stage accordingly as follows. This does not affect general conclusion of our paper.**

At AF stage, exponential regression analysis was carried out with fewer measured $Rs$ values because the duration was shorter than other stages. Due to the fact that active layer became a closed system and that lower number of $Rs$ measurements (two occasions for daily average data) were obtained in the later period of AF, the cumulative $Rs$ (143.74 to 157.34 $gCO_2/m^2$) only accounted for about 8.89% of the total $Rs$ emission in a complete freeze-thaw cycle. It is noteworthy that proportion of respired soil $CO_2$ can be transported via vascular plants, which may function as a conduit for $CO_2$ from deeper soil layers (Ström et al., 2005). Therefore, more frequent observations incorporating vegetation function are warranted to refine the estimated $Rs$ at AF stage proposed in this study.

4) Fig 3: is the main peak (increase) in Rs before the ZC period

**We redrew Figure 3, and, again, the peak of $R_s$ before the ZC period was an artifact of huge observation errors.**

5) For the model evaluation, add root mean squared error analysis.

**Thanks for the suggestion. To evaluate the reliability of the $R_s$ models at different freeze-thaw stages, root mean squared error (*RMSE*) analysis was performed and its results are included.**

6) Which temperature differences used for Q10 estimation?

**To clarify this, we included additional explanation on this. Basically, we calculated $Q_{10}$ values from dataset of each stage, which 
[revised manuscript text omitted]

---

## Author Response (AR2)

Dear Professor Beer,

We are grateful to you for the helpful and constructive suggestions and comments. We have made a major revision to address all issues raised by you, which we believe improved the manuscript substantially. The revised contents were marked with red fonts in the text. We highlight here how we have addressed your suggestions.

1) Fig 3. Please discuss further the overestimation in spring and underestimation in autumn by the (temperature-driven) model.

**We acknowledged that our models have limitation due to low sampling frequency and the limited sampling time of a day due to harsh field conditions and logistic restriction. However, the RMSE values and the similar variation trends between modelled and measured Rs suggest that our model is still valid with the information we gather. We discuss this in the text as:**

At AF stage, exponential regression analysis was carried out with fewer measured $R_s$ values because the duration was shorter than other stages. The modeled $R_s$ fluxes was generally lower than that measured during this stage (Fig.3). These biases between the measured and modeled $R_s$ fluxes were likely to be caused by sampling scheme. The low sampling frequency (two occasions for daily average data) in the period of AF could increase the variance of aggregated estimates (Ryan and Law, 2005). In addition, measurements during this stage were usually restricted to daytime and dry days, and the sampling would inevitably miss the pulse of microbial or root activity immediately following occasional precipitation (Sotta et al., 2004). Thus, the cumulative $R_s$ (143.74 to 157.34 $gCO_2/m^2$) calculated by an exponential model only accounted for about 8.89% of the total $R_s$ emission in a complete freeze-thaw cycle, which probably underestimated the $R_s$ emission during this stage. Although the active layer gradually became a closed system in this stage, it is noteworthy that a proportion of respired soil $CO_2$ can still be transported via vascular plants, which may function as a conduit for $CO_2$ from deeper soil layers (Ström et al., 2005). Furthermore, the diurnal freezing and thawing actions occurring in this stage also played an important role on the $R_s$ emissions (Contosta et al., 2013). Therefore, more frequent observations with automated chambers incorporating vegetation function are warranted to refine the estimated $R_s$ at AF stage in this study.

At SW stage, the modeled $R_s$ fluxes showed a rising trend in the ranges from 0.42 to 0.72 $\mu mol/m^2 s$, and were generally higher than those measured in 2017 (Fig.3). These biases may also be caused by low sampling frequency and simple averaging for daily average data in regression analysis (Ryan and Law, 2005). The diurnal freezing-thawing process during this stage also stimulated the activities of soil microorganisms and promoted the $R_s$ emissions, but the low sampling frequency and the restricted sampling time (between 9:00 and 11:30 a.m. local time) probably missed the peaks and pulses of $R_s$ fluxes of a day. Therefore, the measured $R_s$ flux may underestimate the actual emission rate in this stage. However, it is also reported that biases of chamber-based estimates of $R_s$ can be reduced by using a regression model which is extrapolated with soil temperature and moisture (Ryan and Law, 2005). In addition, the smaller value of *RMSE* at this stage also testified the temperature-driven model was preferable for the $R_s$ prediction (Fig.5). Thus, the cumulative $R_s$ (181.43–198.37 $gCO_2/m^2$) calculated by the

exponential model was estimated to be 11.29±0.11% of the total $R_s$ emission in a complete freeze-thaw cycle. Despite of the possible biases between the modeled and measured $R_s$ fluxes, our model is still a reliable estimate for $R_s$ emission during this stage, which is further supported by the same trends in the variations between the two values. The increasing trend in $R_s$ fluxes can be caused by the following mechanisms: First, the activation of soil respiration was mediated by increased soil microbial activities as soil temperature and water content increased. Furthermore, as spring proceeds with warming of soil, the mobilization of stored carbohydrates enhanced soil respiration (Davidson et al., 2006). Finally, daily freeze-thaw actions in late April may have further enhanced the soil respiration quickly.

2) Fig. 4: I assume the data are observations? Please, clarify in the caption.
   **Yes, and we clarified it in the caption**.
   Fig. 4. Relationship between soil temperature and moisture at 5cm depth and measured $R_s$ flux for the summer thawing stage (ST), autumn freezing stage (AF), winter cooling stage (WC), and spring warming stage (SW)

3) Fig 6 and discussion 4.1.: Please, justify why these model results are important for our understanding of the processes. In Fig 4 we see that the model results are biased in spring and autumn, why are these results now used for understanding the processes and not the observations. For example, in Fig 6, AF stage, observations decrease and are not stable over the time. What do we learn here from the model results that are also only driven by temperature?

**Soil respiration ($R_s$) is the major pathway for carbon exiting terrestrial ecosystems and it has been treated as strictly a heterotrophic process in many models and syntheses with responding to temperature or moisture (Kicklighter et al.1994; Raich and Potter 1995). For the heterotrophic process, soil temperature and moisture are the two important factors to influence the enzyme activities and amount of substrate pool and hence microbial respiration (Pendall et al. 2004). However, for the active layer in permafrost regions, the biggest characteristic of the freeze-thaw process was its complex variations in soil temperature and moisture and the changes of soil temperature and moisture content at different freeze-thaw stages again affected the soil microbial activities, aeration status, and biochemical properties, which regulated the $R_s$. According to the characteristics of soil temperature and water change, the freezing-thawing cycle of active layer can be divided into processes of cooling, start freezing to fully freezing, dropping in temperature, rising in temperature but still in frozen state, start thawing to fully thawing, and rising in temperature but in thawed state (Jiao and Li, 2014). So, we determined the $R_s$, the soil temperature and moisture of the different freezing-thawing processes and used a regression model with soil temperature or moisture to reduce the variance of aggregated estimates under the conditions of harsh environmental conditions and lack of manpower with low sampling frequency. In addition, although biases existed between the modeled and measured $R_s$ fluxes especially in the AF (autumn freezing) and SW (spring warming) processes in the present study, RMSE analysis showed that the exponential models of $R_s$ were preferable for $R_s$ prediction at different freeze-thaw process (RMSE<0.67). Hence, we think these model results are important for our understanding of the processes. For**

the problem of low sampling frequency, resulting in greater biases between the modeled and measurement $R_s$ fluxes, especially in the AF and SW stages, we will take more frequent observations with automated chambers to refine the estimated $R_s$ in the follow-up work.

In Fig.4, we made a regression analysis basing on the measured $R_s$ fluxes and soil temperatures at 5cm depth and have clarified its caption. The reasons why the model results were used for understanding the processes and not the observations, especially when the model results were biased in spring and autumn are: First, due to harsh environmental conditions and lack of manpower especially in winter seasons on the Qinghai-Tibet Plateau, measurements with low sampling frequency were just undertaken, the peaks and pulses of $R_s$ fluxes could have been missed; Second, low sampling frequencies easily increased the variance of aggregated estimates but using a regression model with soil temperature is a preferable method to reduce the variance of aggregated estimates (Ryan and Law, 2005).

In the present study, from the model results that are also only driven by temperature, we learn that although the rates of soil respiration are usually positively correlated to soil temperature, soil moisture can easily become a limiting factor when the soil moisture content is high (for example at the ST stage) during the freezing-thawing processes of active layer. In fact, multiple factors usually interactively affected the soil respiration but we cannot separate their interactions completely. In addition, we also find that the soil respiration is not sensitive to moisture under low temperatures. For example, when the soil temperatures were below 2°C at the SW stage, $R_s$ fluxes didn't have big ups and downs as the soil moisture fluctuated greatly (Fig.6d). This result is well consistent with the study by Luo et al (Luo and Zhou, 2006). Similarly, soil respiration is not very sensitive to temperature under low moisture (below 7%). Among all the different freezing-thawing processes, the AF process maybe is special due to the active layer gradually developing from an open system into a closed one as the surface soil became frozen. The free exchanges of gas between the active layer and atmosphere were blocked. As a result, the soil respiration fluxes were not well correlated with soil temperatures or moistures. So, much more effort is needed to find an indicator to strongly link the soil respiration and the freezing-thawing process of active layer.

[Figure]

**Fig.6d. Variations in modeled soil respiration ($R_s$), soil temperature ($T_s$) and soil water content (SWC) for the SW stage. The SWC unit stands for water volume per total soil volume. The error band of modeled $R_s$ stands for 95% confidence interval.**

Minor: Please, add figure captions a,b,c etc. for clarity.

**We have added a, b, c etc. in the figure captions.**

References:

Kicklighter DW, Melillo JM, Peterjohn WT, et al. Aspects of spatial and temporal aggregation in estimating regional carbon dioxide fluxes from temperate forest soils. Journal of Geophysical Research, 1994, 99(1): 1303-1315.

Raich JW, Potter CS. Global patterns of carbon dioxide emissions from soils. Global Biogeochemical Cycles, 1995, 9(1): 23-36.

Pendall E, Bridgham S, Hanson PJ, et al. Below-ground process responses to elevated $CO_2$ and temperature: a discussion of observations, measurement methods, and models. New Phytologist, 2004, 162: 311-322.

Jiao YL, Li R. Processes of soil thawing-freezing and features of soil moisture migration in the permafrost active layer. Journal of Glaciology and Geocryology, 2014, 36: 237-247.

Ryan MG, Law BE. Interpreting, measuring, and modeling soil respiration. Biogeochemistry, 2005, 73: 3-27.

Sotta ED, Meir P, Malhi Y, et al. Soil $CO_2$ efflux in a tropical forest in the central Amazon. Global Change Biology, 2004, 10: 601-617.

Contosta AR, Frey SD, Ollinger SV, et al. Soil respiration does not acclimatize to warmer temperatures when modeled over seasonal timescales. Biogeochemistry, 2013,112: 555-570.

Luo YQ, Zhou XH. Soil Respiration and the Environment, Chapter 5: Controlling factors. 2006, Pages 79-105.